# Applying artificial snowfalls to reduce the melting of the Muz Taw Glacier, Sawir Mountains

Feiteng Wang[1*], Xiaoying Yue[1], Lin Wang[1], Huilin Li[1], Zhencai Du[2], Jing Ming[3], Zhongqin Li[1]

1       State Key Laboratory of Cryospheric Science / Tien Shan Glaciological Station, Northwest Institute of Eco-Environment and Resources, Chinese Academy of Sciences, Lanzhou 730000, China
2       Center for Monsoon System Research, Institute of Atmospheric Physics, Chinese Academy of Sciences, Beijing 100029, China
3       Beacon Science & Consulting, Doncaster East, VIC 3109, Australia

**Correspondence**
* Feiteng Wang, wangfeiteng@lzb.ac.cn

**ABSTRACT**

The glaciers in the Sawir Mountains, Altai area, have been experiencing a continuing and accelerating ice loss since 1959, although the snowfall here is abundant and evenly distributed over the year. As an attempt to reduce their melting, we carried out two artificial-snowfall experiments to the Muz Taw glacier during 19 – 22 Aug 2018. We measured the albedo and mass balance at different sites along the glacier before and after the experiments. Two automatic weather stations (AWS) were set up at the equilibrium line altitude (ELA) of the glacier as the target area and the forefield as the control area to record the precipitations, respectively. The comparison of the two precipitation records from the two AWSs suggests that natural precipitation could account for up to 21% of the snowfall received by the glacier during the experiments. Because of the snowfalls, the glacier's surface albedo significantly increased in the mid-upper part; the average mass loss during Aug 18 – 24 (after the experiments) decreased by between 32 (14%) and 41 (17%) mm w.e. comparing to that during Aug 12 – 18 (before the experiments); and the mass resulting from the snowfalls accounted for between 42% and 54% of the total melt during Aug 18 – 24. We also propose a mechanism involving artificial snowfall, albedo and mass balance and the feedbacks, describing the role of snowfall in reducing the melting of the glacier. The work in current status is primitive as a preliminary trial, the conclusions of which need more controlling experiments to validate in larger spatial and temporal scales in future.

**Keywords**
artificial snowfall, Muz Taw Glacier, Sawir Mountains, glacier mass balance, reduce melting

## 1 Introduction

Mountain glaciers are an essential part of the cryosphere. As high-altitude reservoirs, they are vital solid-water resources (Immerzeel et al., 2019; Immerzeel et al., 2010). Glacier fluctuations represent an integration of changes in the mass and energy balance and are well recognized as high-confidence indicators of climate change (Bojinski et al., 2014). Satellite and in-situ observations of changes in the glacial area, length and mass show a global coherence of continued mountain-glacier recession in the last three decades with only a few exceptions (Zemp et al., 2019). For the Sawir Mountains, the ablation of the glaciers is more intense than the global average, and the total area of the glaciers reduced by 46% from 23 $km^2$ in 1977 to 12.5 $km^2$ in 2017 (Wang et al., 2019). The accelerated retreat of glaciers not only causes spatial and temporal changes in water resources but also has a significant impact on sea-level rise, regional water cycles, ecosystems and socio-economic systems (such as agriculture, hydropower and tourism); the melting of glaciers also increases the occurrence of glacial disasters, such as glacial lake outburst flooding, icefalls and glacial debris flows (Hock et al., 2019).

So far, there are not so many approaches used in practice for reducing the rate of glacier ablation. Some technical measures, including energy conservation, temperature-increase control and establishing glacial reserves, have been taken to reduce the ice melting on Earth. In recent years, new ideas and techniques have emerged for slowing the melting of glaciers. For example, in the Rhone glacier of the Swiss Alps, white blankets are used to shelter the glacier and slow down its melting (Dyer, 2019). In the Morteratsch Glacier of the Alps, artificial snow was expected to be applied for slowing down the glacier melting (Oerlemans et al., 2017). In Austrian glacier ski resorts, over 20-m thickness of the ice was preserved on mass balance managed areas compared to non-maintained areas during 1997 – 2006 (Fischer et al., 2016).

Cloud-seeding over a glacier to generate and enhance snowfall for reducing mass loss has rarely been tested in previous study. There have always been controversial discussions on the virtual efficacy and positiveness of using AgI smoke to seed cloud and enhancing precipitations since the measure was introduced by Vonnegut (1947). The controversy mainly resides in two sides. One side claimed that no statistical or

physical evidence had been provided to establish the scientific validity of the
operations (Council, 2003; Silverman, 2001), while the other affirmed that the past
operations conducted in Australia successfully increased precipitations by 5% up to
50% (Bowen, 1952; CSIRO, 1978; Smith, 1967). However, both sides agreed that
the experiments of seeding clouds and producing more precipitation were promising
and deserved more observations to understand the link of physical reactions leading
to precipitation (Council, 2003). A peer review report on global artificial-snowfall
activities by the World Meteorological Organization suggests that the toxicity of the
seeding material (majorly silver iodine, i.e. AgI) is unlikely to trigger environmental
hazards (Flossmann et al., 2018). A potential concern is that artificial-precipitation
activities might redistribute the natural precipitation over a region; however, applying
cloud seeding over the mountain glaciers usually up to 5 km in length in Central
Asia, is presumably acceptable.

As an attempt, we select the Muz Tau glacier in the Sawir Mountains as the
projected glacier. During the glacier's ablation period in 2018, we tried to induce
artificial precipitations by using the ground AgI smoke generators to seed clouds
over the glacier. These smog generators were set up there by the local
meteorological service for artificial-precipitation tasks. We also combined the
precipitation amounts and type, time and frequency recorded by the rainfall gauge
and the mass balance and albedo of the glacier measured to study the role of
artificial snowfall in reducing the mass loss of the glacier.

**2  The Sawir Mountains and the Muz Taw Glacier**
The Sawir Mountains span the border shared by China and Kazakhstan and are the
transitional section between the Tianshan Mountains and the central Altay
Mountains. The Muz Taw Glacier (47°04′N, 85°34′E) is a northeast-orientated valley
glacier with an area of 3.13 $km^2$ and a length of 3.2 km in 2016, located on the
northern side of the Sawir Mountains (Figure 1). Its elevation from the terminus to
the highest point ranges from 3137 m to 3818 m a.s.l. and its ice volume is 0.28 $km^3$,
with an average ice thickness of 66 m (Wang et al., 2018).

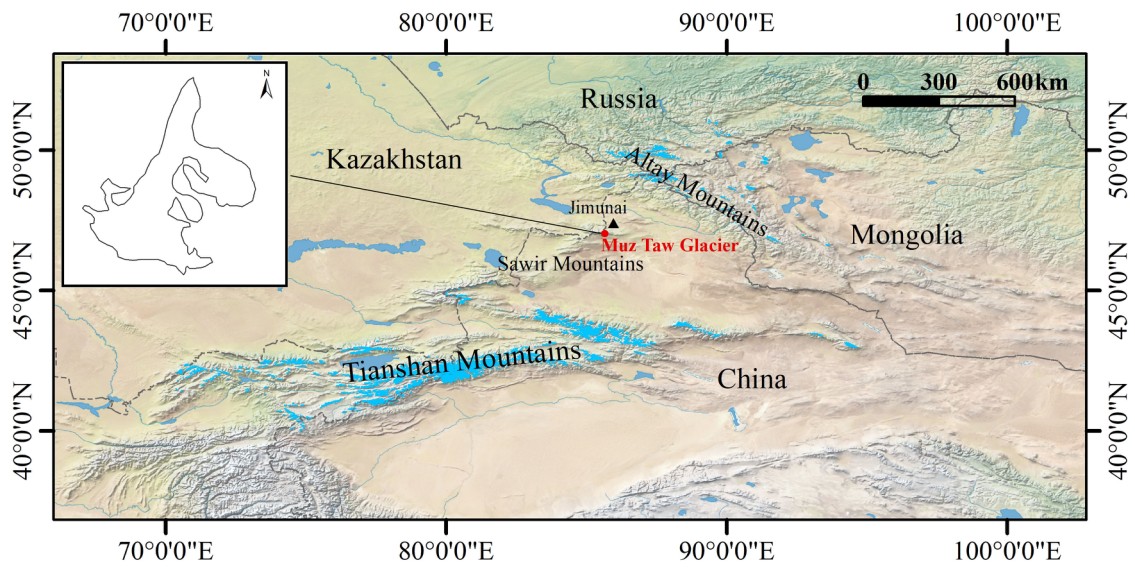

*Figure 1 Location of the Muz Taw glacier and the Sawir Mountains, where the map in the background is downloaded from the website https://www.naturalearthdata.com/ and the outline of the glacier is sourced in Guo et al. (2015).*

The general circulation over the study area is featured by the prevailing westerlies interacting with the Asian anticyclone and polar air mass in winter (Panagiotopoulos et al., 2005). At the Jimunai Meteorological Station (984 m a.s.l.), 46 km northeast of the Muz Taw Glacier, the annual mean air temperature measured was 4.27 °C; the annual mean precipitation was 212 mm during 1961–2016, and the winter precipitation accounted for 10% - 30% of the annual total.

The Muz Taw Glacier has been in constant recession since 1959 (Wang et al., 2019). Especially for the past 20 years, it has been experiencing a rapid and accelerated shrinkage. From 1977 to 2017, the glacier area decreased by 10.51 km$^2$, accounting for 45.72 % of its previous surface area (Wang et al., 2019). The average retreat rate of the glacier terminus was 11.5 m a$^{-1}$ during 1989 – 2017. The latest measurements show the mass balance of the Muz Taw Glacier was – 975 mm w.e. in 2016, – 1192 mm w.e. in 2017 and – 1286 mm w.e. in 2018, respectively; and the annual equilibrium line of the glacier was approximately 3400 m a.s.l. (Song, 2019).

## 3  Field Experiments and measurements

### 3.1  Meteorological radar observations

We used a WR-08X digital radar system (Wuxi Leyoung Electronics Technology Co., Ltd) built up at the Jimunai Meteorological station to identify the precipitation clouds

around the Sawir Mountains. The radar is a new X-band digital weather radar
capable of detecting meteorological targets within 300 km. The radar can
quantitatively detect the spatial distribution of intensity of cloud rain targets below 20
km distanced from 5 km to 150 km and their motions (e.g., developing height,
moving direction and speed.). It can also provide real-time meteorological
information. A more detailed description of its application in this area can be referred
to in Xu et al. (2017).

**3.2   Artificial-precipitation experiment**
The Muz Taw glacier is developing along the valley, and the terminal is the heading
source of the Ulequin Urastu River and Ulast River. Fourteen silver-iodide (AgI)
smog generators have been distributed along the rivers for artificial-precipitation
tasks by the local meteorological service. These smog generators use solar power to
light and are remotely controlled. The AgI sticks used in the generators allow to
generate $10^{14}$ AgI-contained ice nuclei per gram at $-7.5\ °C \sim -20\ °C$ (Kong et al.,
2016). In the daytime, valley winds prevail along the valley up to the glacier due to
intense radiation and the heating-and-lifting effect for air over the snow surface. It is
ideal for generating AgI smogs and carrying them by the upwards air stream over the
glacier surface to form precipitations. No extra water is needed to form precipitations
in our experiments. We monitored the distribution and structural developing of clouds
and identified the orientation, height and distance of the clouds approaching the
glacier at the radar station. Associated with observing the moving of the potential
target clouds and the receiving of the reflection of the radar transmission, we ignited
the smog generators for seeding artificial precipitations, when we realized the
possibility is high enough to form precipitation potentially (Figure 2). The detailed
operation of conducting artificial precipitations in the studied glacier has been
described in Xu et al. (2017).

First, we used the radar to identify local convective clouds in the background
synoptic clouds and measured the orientation, height and distance of the
convections for determining the time and area for performing artificial precipitation
seeding. And then we chose most favourable timing to ignite the silver-iodide smog
generators (Figure 3a) and let the silver-iodide (AgI) particles as catalyzer help
forming amounts of artificial ice nuclei (Figure 3b) to absorb more water vapour and
promote to form precipitations.

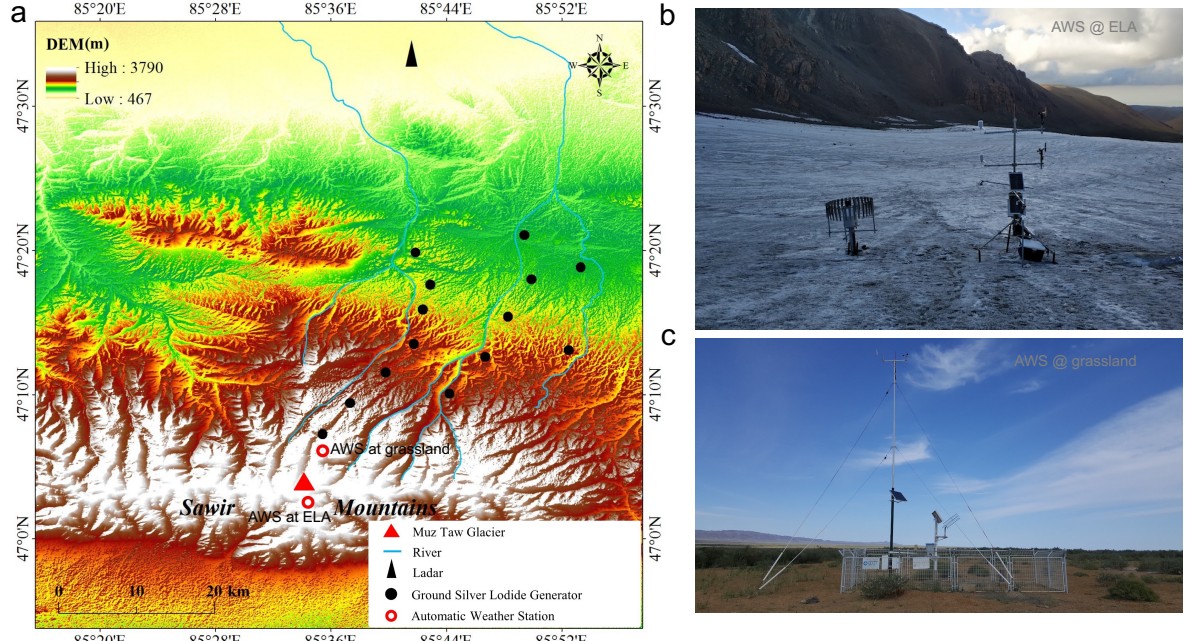


*Figure 2 a) The map of the study area, including the Muz Taw glacier, the two automatic weather*
*stations (AWS) set up at the equilibrium line elevation (ELA) and the forefield of the glacier and the*
*distribution of the silver-iodide-smog generators along the Ulequin Urastu River and Ulast River in the*
*Sawir Mountains for seeding artificial precipitations, b) the AWS set up at ELA and c) the AWS set up*
*at grassland with a straight distance of ~5 km north to the AWS at ELA.*

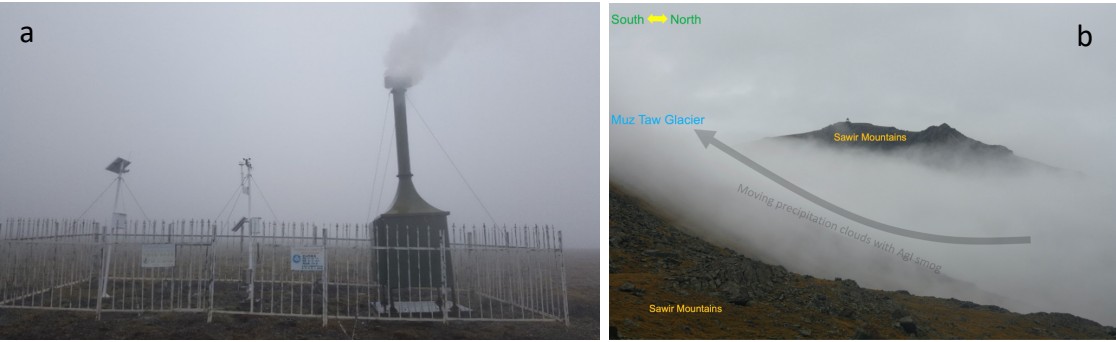


*Figure 3 a) Igniting the AgI smog generators along the terminal river when the cloud accumulated late*
*on the afternoon of 19 and 22 Aug 2018, and b) the accumulating of clouds in the valley of the Muz*
*Taw Glacier favoured by the AgI particles moved up towards the summit of the glacier.*

## 3.3    Measurement by the automatic weather stations (AWS)

We set up an automatic weather station (AWS at ELA) on a relatively flat surface
near the equilibrium line altitude (ELA) of the Muz Taw glacier since 8 Aug 2018 (47°
03′36″N, 85°33′43″E, 3430 m a. s. l.; Figure 2a&b and Figure 4). The AWS has
various sensors to fulfil the requirement of our study (Table 1). A thermometer
(Pt100 RTD, ± 0.1 K) was mounted horizontally 1.5 m above the surface to measure
air temperature. The measurement of albedo was calculated by measuring incoming
and reflected shortwave radiation with the CNR4 pyranometer mounted on the AWS
at the height of 1.5 m. The error of pyranometer is smaller than 1% in the wavelength
from 0.3 µm to 2.8 µm. Precipitation was measured by an auto-weighing gauge (T-
200B, Geonor Inc.) with an accuracy of about ± 0.1%. All sensors were connected to
a data logger (CR6, Campbell) which is able to work in low temperature (-55 °C) and
record the hourly means every ten seconds. In the forefield of the glacier around five
kilometers north of the AWS at the ELA, another AWS on the grassland (AWS at
grassland) was set up by the local meteorological service to monitor conventional
meteorology (Figure 2a&c).

*Table 1 The sensors mounted on the AWS at ELA and their technic features*

| Sensor | Measurement | Model | Accuracy or features |
|---|---|---|---|
| Thermometer | temperature | Pt100 RTC | ± 0.1 K |
| Pyranometer | radiation | CNR4 | < 1% in 0.3 - 2.8 µm |
| Auto-weighing gauge | precipitation | T-200B, Geonor Inc. | ± 0.1% |
| Data logger | data recording | CR6, Campbell | working in low temperature |


**3.4   Measurement of the surface spectral reflectance**
We used an ASD Fieldspec HandHeld 2 Spectroradiometer to measure the
reflectance data at 325-1075 nm by with a resolution of 3 nm and an error of less
than 4%. The measurement sensor fitted with a bare fibre was mounted on a tripod
at 0.5 m above the surface and had a 25° field of view to a spot sized ~0.225 m in
diameter. The spectroradiometer was calibrated to hemispherical atmospheric
conditions at the time, by viewing white-reference panel and then viewing the glacier
surface. We recalibrated the instrument on occasion when the sky radiation
conditions changed. To minimize the influence of slope and solar zenith angle on
albedo, we conducted the measurements in a water-level plane within 12:00-16:00
local time. At each sampling site, three consecutive spectra consisting of ten dark
currents per scan and ten white reference measurements were recorded and
averaged. Meanwhile, cloud cover and surface type were noted for each
measurement.

We measured spectral reflectance at fourteen sites across the glacier, on 18, 20, 22
and 24 Aug 2018 (Figure 4). In house, the spectrum data were exported from the
instrument by the Spectral Analysis and Management System software (HH2 Sync).
The broadband albedo was calculated as a weighted average based on the spectral
reflectance and the incoming solar radiation across the entire spectral wavelengths
at each site (Ming et al., 2016; Moustafa et al., 2015; Wright et al., 2014; Yue et al.,
2017). The period-mean albedo averaged for the 14 sites before and after
conducting artificial-precipitation experiments (12 – 18 Aug and 18 – 24 Aug) are
shown in Table 2. We excluded the apparent outliers (higher than 0.98) of the albedo
data which are physically unrealistic.

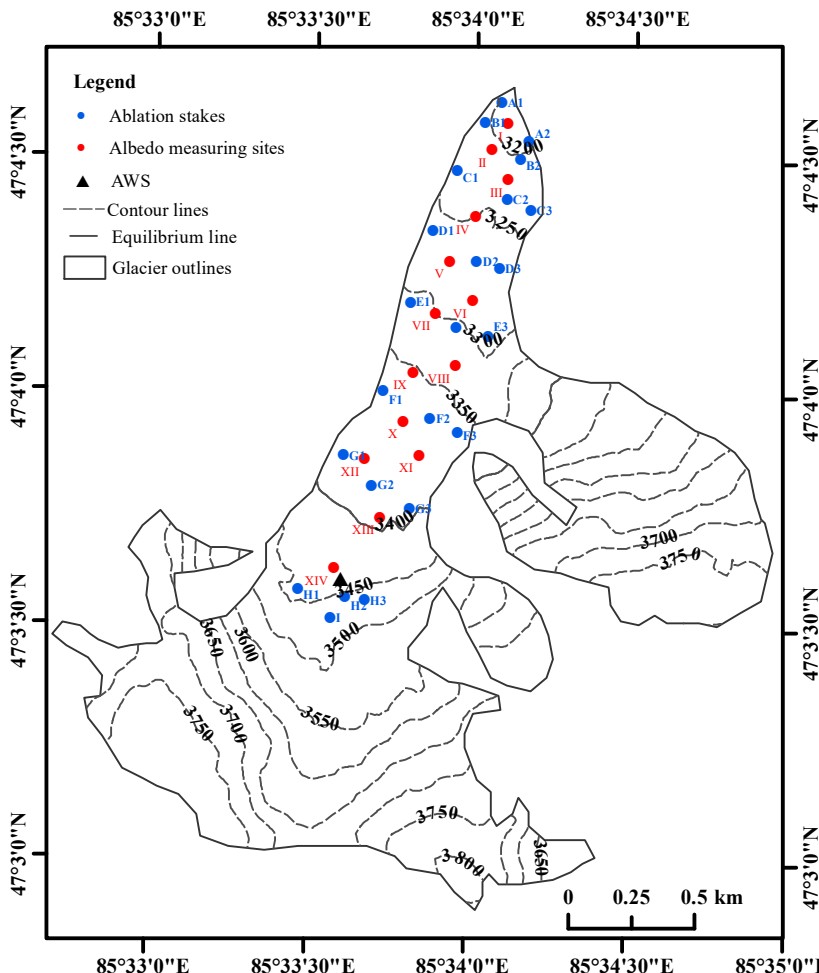

*Figure 4 The location of the AWS at ELA and the measuring sites for surface albedo and mass*
*balance on the Muz Taw glacier.*

**3.5   Measurement of the mass balance**
We have measured the mass balance of the Muz Taw Glacier annually since 2014
with the method introduced in Østrem and Brugman (1991). Metal stakes for mass-
balance measurements were fixed into the ice with a portable steam drill. The stake
network consisted of 23 stakes evenly distributed in different altitudes, where three
stakes in every row roughly (Figure 4). The stick scale for measuring balance was
read thrice, 12, 18 and 24 Aug, respectively. We compared the mass varying
between the two periods (12-18 Aug and 18-24 Aug). The snow depth at each stake
was measured by reading the scale, and the density of snow was measured by
weighing the mass of snow with a given volume. We used the depth and density
data of snow to calculate the mass balance at the stake sites. The mass balance
was obtained on 1 May and 31 Aug annually. For verifying the effect of artificial
snowfalls on the mass balance of the glacier, in particular, we conducted three
additional measurements for the mass balance on 12, 18 and 24 Aug 2018,
respectively. The baseline of all the mass balance data in this study is the mass
balance measured by the stakes on 12 Aug. The calculation of the mass balance of
the whole glacier is following an interpolated method based on singular-point
measurements introduced by Wang et al. (2014).

**4  Results and discussion**
**4.1   Natural or artificial precipitations and their amounts and forms**
Figure 5a shows the hourly temperature and precipitations recorded by the AWS at
ELA from 12 to 24 Aug 2018. There were some natural precipitations during 12 – 14
Aug, while except this and that in the experiment days, the whole period of 12 – 24
Aug was sparse in precipitations. Artificial-precipitation experiments were carried out
on 19, 22 and 23 Aug. The precipitation rates on 19th, 20th, 22nd and 23rd were 6.2,
1.3, 1.8 and 10.6 mm/day, respectively. Most snowfalls were observed during
midnights and early mornings. It seems not likely to distinguish the artificial
precipitations from the natural ones if they were simultaneously mixed in all these
events.

Previous weather modification experiments using the same method as ours
concluded that it was challenging to tell that how much artificial precipitation mixed in
the whole amount directly came after conducting the cloud seeding (CSIRO, 1978;
Qiu and Cressey, 2008; Ryan and King, 1997). The results from the measurements
by Marcolli et al. (2016) and (Fisher et al., 2018) suggested the efficacy and success
of using AgI on growing ice nuclei in clouds and promoting snowfall. In our study,
there were significant precipitation amounts recorded by the AWS at ELA every
single time after we ignited the smoke generators, associated with a highly
significant linear relationship (n = 10, $r^2$ = 0.9999) between the timings of igniting AgI
and recording snowfalls (Figure 5b). The co-occurring of the significant snow falling
using the AgI smoke to seed cloud (Figure 3 and Figure 5b) allows supposing that
we were producing artificial precipitations.

The AWS at grassland in the forefield of the Muz Taw glacier was clear from the AgI
smoke during the AP experiments. This allows us to use the precipitation data
recorded by it as a control to distinguish natural precipitation from the artificial
recorded by the AWS at ELA. We lost the precipitation data from the AWS at
grassland during the first AP experiment on Aug 19 for the rain gauge was full and
overflowed. While for the second experiment, the precipitation data were
successfully collected from the AWS at grassland for a comparison.

Figure 6 shows the precipitations recorded by both AWSs and the record ratio of
grassland to ELA during the second AP experiment (Aug 22 to 23). The
precipitations recorded by the two AWSs were not synchronized. The AWS at ELA
did not record any precipitations when that at grassland recorded at 19:00 and 20:00
on Aug 22; while there were records after 6:00 by the AWS at ELA but none for at
grassland. The correlation between the two precipitation records is fairly weak ($r^2$ =
0.05) when they were both recorded by the AWSs, implying that the cause of
precipitation (i.e. natural or artificial) might be distinctly different or likely mixed on
the target area.

We presume two possibilities of whether there was natural precipitation joined the
artificial process targeted on the glacier. The first was none natural precipitation took
part in conceiving artificial snow on the target area, and the second, if any, was a
part of this. The ratios of precipitations by the AWS at grassland to that at ELA were
smaller than 35% with a mean of 21 ± 3 % (Figure 6), which could be used for
estimating how much naturally induced precipitation taking part in the AP experiment
based on the second presumption.

To determine the amount of solid precipitations that accumulates on the glacier
surface, we apply a sinusoidal function (Möller et al., 2007) on the total precipitation.
The function describes the transition between solid and liquid precipitations in a
temperature range between +2 °C and +4 °C (Fujita and Ageta, 2000; Mölg et al.,
2012). When the air temperature is lower than 2 °C, solid precipitations (snow) will
occur, and between 2 – 4 °C rain would fall with snow. During our experiments, the
air temperatures were below 2 °C when precipitations occur, implying that the
precipitations in the two experiments were solid.

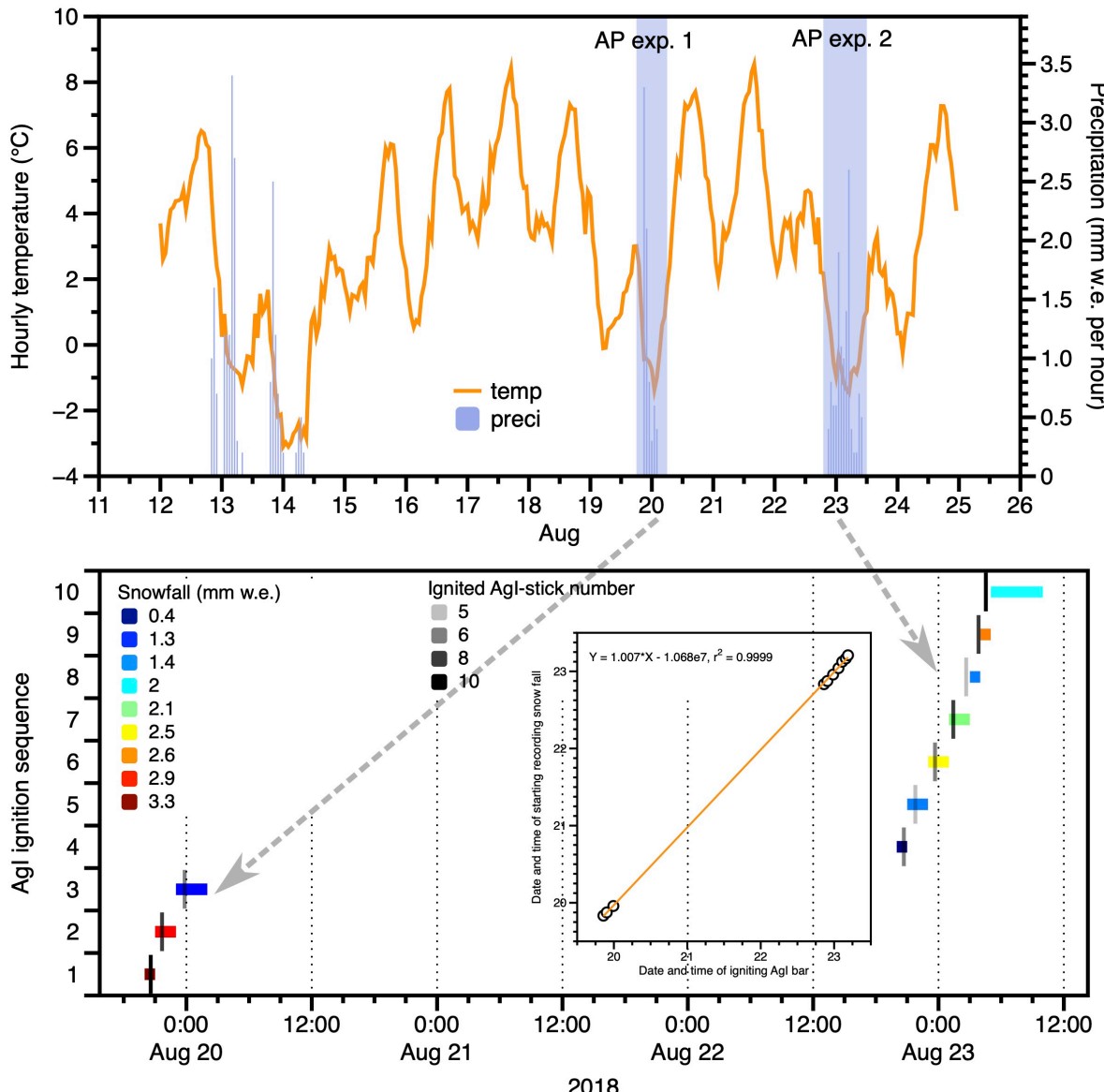


*Figure 5 a) The daily snowfalls and hourly averaged temperature recorded by the AWS at ELA from*
*12 to 24 Aug 2018, where the two artificial-precipitation experiments (AP exp. 1 and 2) are marked,*
*and b) the hourly snowfall amounts (indicated by color) and time periods (indicated by length)*
*recorded by the AWS at ELA and the ignited AgI-stick number (indicated by color) and time during the*
*two experiments.*

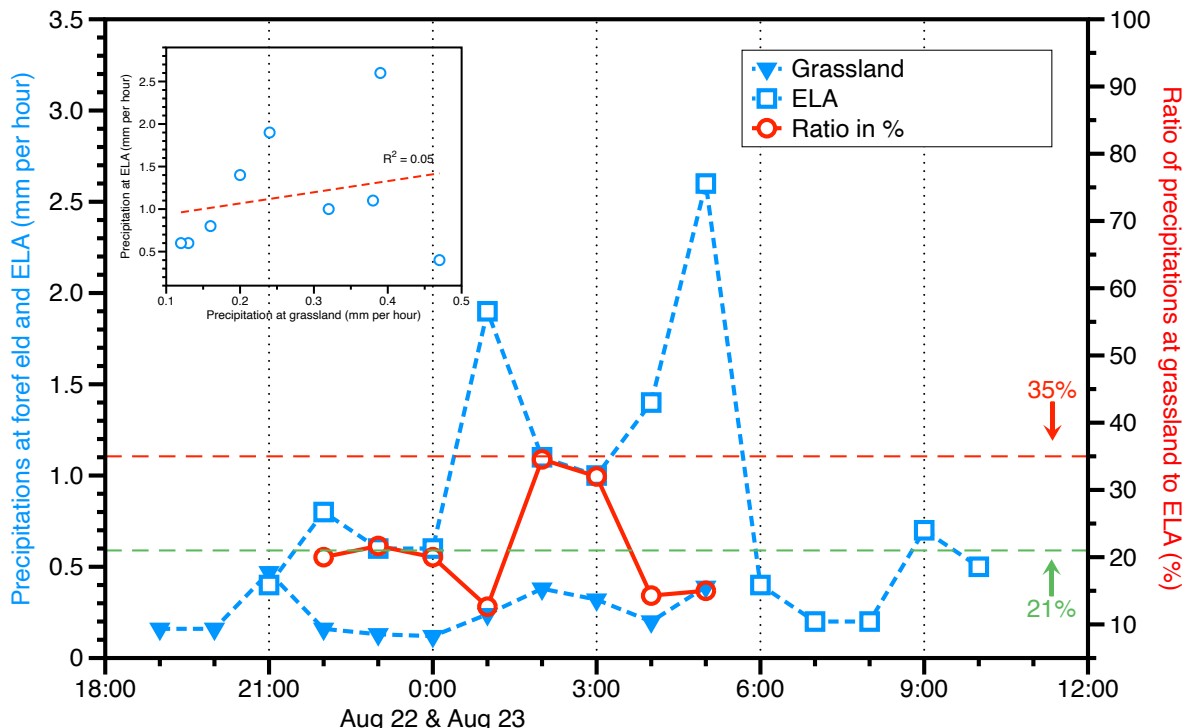


*Figure 6 The precipitations recorded by the AWSs at grassland (inversed blue solid triangles) and*
*ELA (hollow blue squares) and the precipitation-record ratio of grassland to ELA (hollow red circles)*
*during Aug 22 to 23, in which the scatter plot of the precipitations by both AWSs is included and the*
*green and red dashed lines indicate the upper limit and mean of the ratios.*

## 4.2 The effects of artificial snowfall on surface albedo

Glacier albedo is highly sensitive to snowfall. Once a snowfall occurs, it will quickly
whiten the surface of the glacier and increase the albedo. Figure 7 shows the
surface albedo of the Muz Taw Glacier at different locations before and after the
artificial-snowfall experiments. We observed that the surface albedo at the sites
varied from relative flatness (e.g., at site I and site III) to more significant fluctuations
(e.g., at site XII and site VII) between 18 and 24 Aug.

Below 3250 m, the surface albedo (at sites I, II, III and IV) was generally smaller than
0.4 (typical albedo of ice with debris) with mild fluctuations as shown in Figure 7.
From 3250 to 3350 m a.s.l. (at sites V, VI, VII and VIII), significant variations in
albedo were observed, ranging from 0.2 to 0.6. In the area of 3350-3400 m a.s.l.,
more significant variations in albedo were observed between 0.1 and 0.7. Because
this area was located near the equilibrium line, it was highly sensitive to air
temperature and snowfall. Artificial snowfall frequently transited the surface from ice

to snow, and air temperature turned the surface inversely from snow to ice, and thus

dramatic changes in albedo occurred. At sites XIII and XIV, which are much higher

than the equilibrium line, the overall albedo exceeded 0.4 and rose up to 0.8. We

observed a slightly increasing trend in albedo at these two sites (XIII and XIV),

suggesting that the surface was covered by relatively lasting snow owing to artificial

snowfalls.

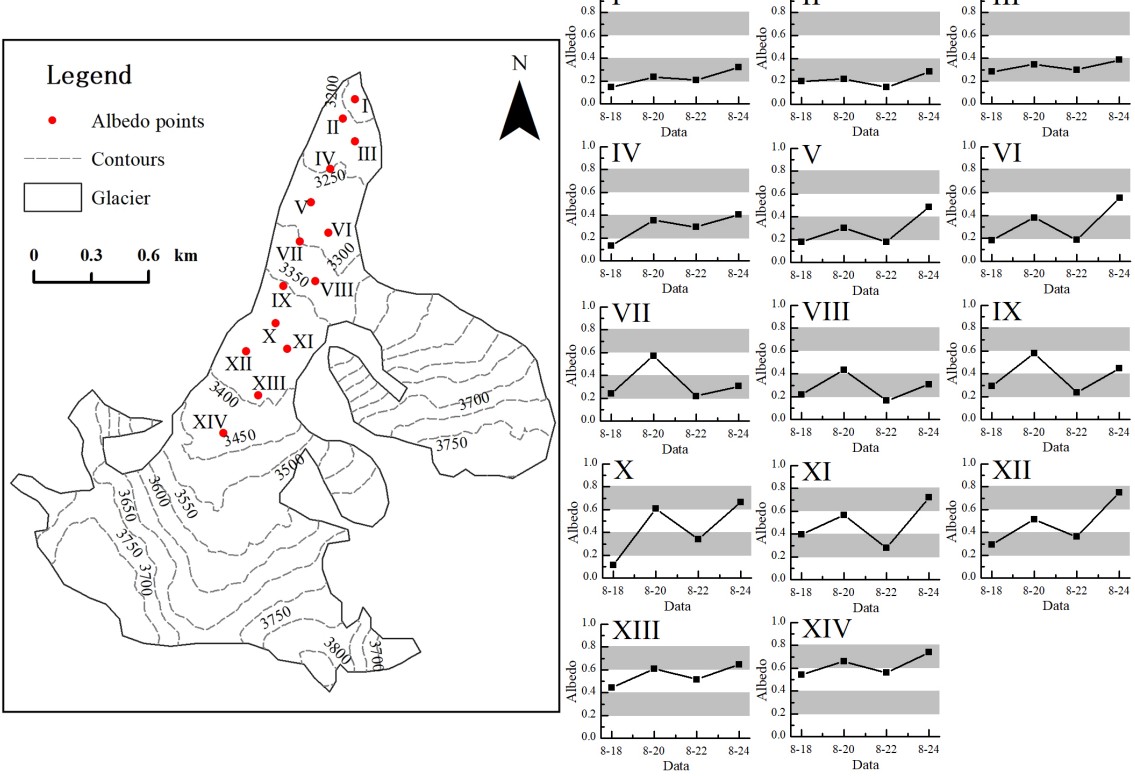

Figure 7 The surface albedo at the fourteen sites (I - XIV) of the Muz Taw Glacier, where the red

points denote the sites and the top-left chart as the reference of the fourteen charts (site I to XIV)

marks the albedo scale and date with the highlighted grey shades.

## 4.3   The varying mass balance responding to the artificial snowfalls

As mentioned in Section 3.4, the stick scale for measuring balance was read thrice

at each site, on 12, 18 and 24 Aug, respectively. To study the effects of the artificial

snowfalls on the mass balance of the glacier, we calculated the mass balance

measured by the stakes during the two periods, i.e. before the artificial snowfalls (12

– 18 Aug) and after the artificial snowfalls (18 – 24 Aug), respectively. The stakes in

a group (A to I) were roughly along the altitude contour (Figure 4), and the

correspondingly measured mass balance of the same group was averaged (Figure

8). The mass balance decrease with altitude from approx. – 400 mm w.e. at 3100 m

to approx. – 100 mm w.e. at the equilibrium line measured by the stakes before the
artificial snowfalls, and decrease from approx. – 300 mm w.e. at 3100 m to approx. –
100 mm w.e. at the equilibrium line after the artificial snowfalls. The difference of the
mass balances measured at the sites between the two periods was 41 ± 15 mm w.e.
averaged on the stake measurements for the Muz Taw Glacier, considering the
difference was completely due to artificial precipitation. If we take 21% of the
difference was due to natural precipitation (Figure 6), the difference would be 32 mm
w.e. Therefore, the difference resulting from the artificial snowfalls accounted for
between 14% (with 21% natural) and 17% (without natural) of the mass balance
before the artificial snowfalls (- 237 mm w.e.).

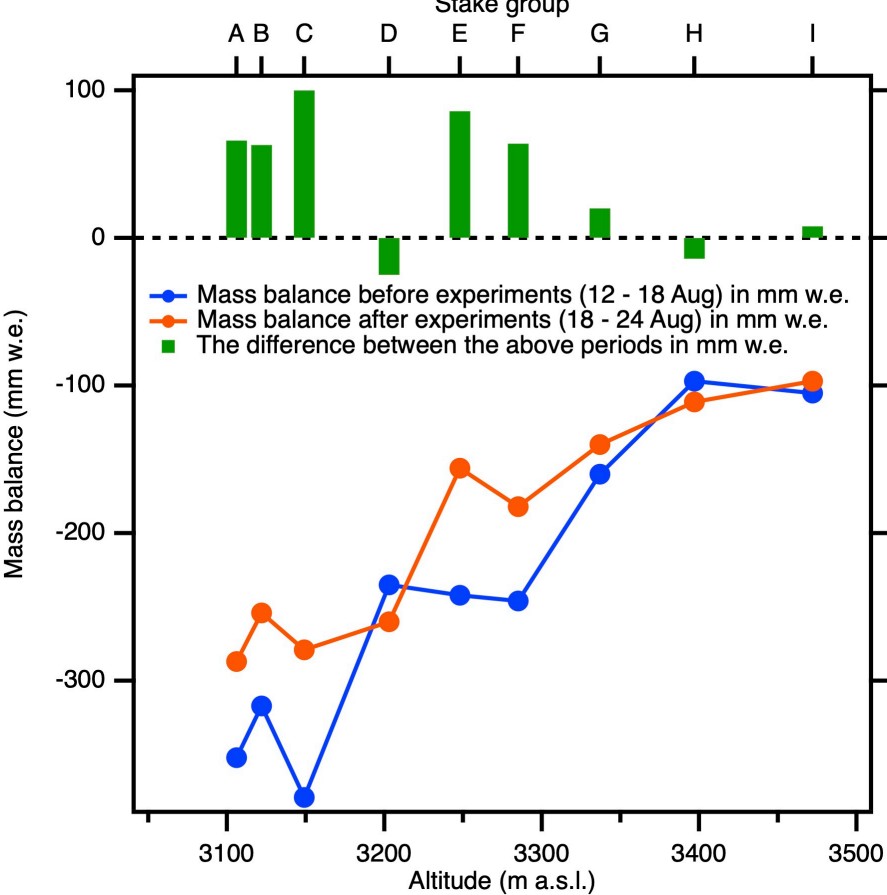

*Figure 8 The averaged mass balance measured at the sites (Stake A - I) before (blue) and after*
*(orange) the artificial snowfalls on 18 and 20 Aug compared with that on 12 Aug (The zero line), and*
*the gained mass (green = orange - blue) due to the artificial snowfalls.*

We compare the positively accumulative temperatures (in brief PAT = $\sum_{i=1}^{n} T_i$, $n$ is
the number of days, and $T$ is the daily averaged temperature in °C), the amounts of
snowfalls, and the surface albedo of the measurements from 12 to 18 Aug ($t_1$) and

from 18 to 24 Aug ($t_2$) (Table 2), respectively. The two periods represent the same time-length span before and after the artificial snowfalls, respectively. The temperature, snowfall and albedo data in this comparison are all from the measurements of the AWS at ELA. The estimated mass balance after interpolating the measured mass balance by the stakes to the whole glacier during $t_1$ and $t_2$ were − 61.4 mm w.e. and − 37.2 mm w.e., respectively. Although the PAT was higher during $t_2$ than during $t_1$, the mass loss of the glacier was 40% lower than $t_1$. More snowfall and higher albedo resulting from the artificial snowfalls can explain the lower mass loss during $t_2$.

*Table 2 The positive accumulated temperatures, snowfalls and albedo measured by the instruments on the AWS at ELA, and the calculated mass balances of the Muz Taw glacier before and after the two artificial-snowfall experiments ($t_1$ = 12 − 18 Aug, and $t_2$ = 18 − 24 Aug).*

| Period | Positively accumulated temperature (°C) | Snowfall (mm w.e.) | Albedo | Mass balance (mm w.e.) |
|--------|------------------------------------------|--------------------|--------|-------------------------|
| $t_1$  | 17.0 | 17.4 | 0.24 | - 61.4 |
| $t_2$  | 18.2 | 19.9 | 0.33 | - 37.2 |

The accumulation at the equilibrium line altitude (ELA) of a glacier is approximately equal to the area average of accumulation over the whole glacier (Braithwaite, 2008). We can presume that the snowfall amount measured by the AWS at ELA of the Muz Taw glacier during $t_2$ was the average received mass of the whole glacier after implementing the AP experiments. The extra melt amount from the glacier besides the gained mass during $t_2$ would be the difference between the calculated mass loss (37.2 mm w.e.) and the snow mass measured by the AWS at ELA (19.9 mm. w.e.), and that would be 17.3 mm w.e. The artificial snowfalls may significantly save the melt of the glacier by 54% during $t_2$, calculated as the percentage of the snowfall divided by the estimated mass balance. Excluding 21% of the mass measured by the AWS at ELA presumably as the contribution of natural precipitations, we conclude that the artificial precipitations buffered the total melting during $t_2$ by 42%.

**4.4    The mechanism: how artificial snowfalls reduce the melting of a glacier**

In the air temperature lower than 2 °C, the artificial snowfall promotes the form of
snow which directly adds mass onto the glacier and increases the mass balance of
the glacier and thereby albedo; the snow cools the surface and increases the surface
albedo; the increased albedo will decrease the solar radiation absorption in the
surface and favour retaining the mass which will, in turn, save the albedo; and
eventually the whole process forms a positive feedback.

This is a very preliminary theory based on the limited data derived from the short-
term experiments, and we need further studies to validate the theory. The albedo
decay of artificial snowfall and snow physics are required to claim a long-term impact
on the mass balance of glaciers. Particularly, the variation in the likelihood of a
snowfall event occurring with or without smoke generators and the partition of natural
and artificial precipitations need to be quantified more confidently, for which more
controlling experiments are needed in future.

**5 Conclusions**
We used AgI-smoke generators to induce artificial snow on the Muz Taw Glacier in
Sawir Mountains on 19 and 22 Aug 2018. Two AWSs were set up on the target
glacier and control area, respectively. The albedo and mass balance were measured
at the stakes evenly distributed along the altitude contours of the glacier before and
after the artificial snowfall experiments. The glacier received a total snow amount of
~ 20 mm w.e. by two experiments, which increased the surface albedo of the glacier.
Larger fluctuations in albedo were measured at the higher sites than lower.

By comparing the precipitations measured by the two AWSs, we conclude that
artificially induced snow could account for at least 79% of the total snow measured
by the AWS at ELA. After interpolating the mass balance measured by the stakes to
the whole glacier, we get a mass balance of – 61 mm w.e. for the period of 12 – 18
Aug and – 37 mm w.e. for the period of 18 – 24 Aug, respectively. The artificial snow
reduced the mass loss of the glacier by ~ 40% due to more snowfall and higher
albedo, nevertheless the positively accumulated temperature during the latter period
was higher than the former.

We compared the mass balances directly calculated from the measurements of the
stakes before the experiments (12 – 18 Aug) with that after (18 – 24 Aug). The
difference between the two periods was between 32 and 41 mm w.e., taking possible
natural snow into account. This suggests that artificial snow does add mass to the
glacier, which is consistent with the result by interpolating stake measurements to
the whole glacier. We also compared the total melt of the glacier during 18 – 24 Aug
with the artificial snow received by the glacier, implying that artificial snow
significantly saved the mass loss by between 42% and 54% after the experiments.

We propose a theory describing the role of snowfall in reducing the melting of the
glacier. The mechanism determines that the environmental temperature and the form
of snowfall, and clouds are the two main factors resulting in the mass gain and loss
of a glacier. Mechanical erosion, energy exchange (thermal-dynamic) and albedo-
induced radiation absorption play major roles in the process of mass varying. This
hypothesized mechanism is preliminary and needs more measurements to
consolidate.

The approach in our work uses solar power to ignite the seeding material for forming
clouds and uses no extra water but redistributes natural water in the local
atmosphere at a small spatial scale. The energy-and-water saving techniques of the
approach with reasonably mass-loss-reducing efficiency from the Muz Taw glacier
validates its efficiency to possibly be applied in more Central-Asian glaciers to
reduce their rapid melting. Especially in summer when the melting is dramatic in the
Central-Asian glaciers, applying the approach suggested by our study on a much
broader scale might reduce the melting significantly. However, it is important to note
that our approach needs a priori atmospheric conditions favourable to precipitation
and can not be applied if the weather is dry and sunny. This study is preliminary and
short in operating time and needs more sophisticated experiments at control and
target areas to partition natural and artificial precipitations. The approach would
sophisticate itself when being implemented more regularly in future repeated and
longer-term, or scaled-up experiments.

**Code/Data availability**
It is currently shared by communities that the dataset would be publicly available
upon acceptance of publication. Please directly contact the corresponding author F.
Wang (wangfeiteng@lzb.ac.cn) or the coordinating author J. Ming
(petermingjing@hotmail.com) for the data repository and the authors will response
accordingly.

**Author contributions**
F.W. conceived the main ideas, designed the experiment and drafted the
manuscript. X.Y., L.W., H.L. and Z.D. helped to design the experiment and collect
the data. J.M. reanalyzed the data and plots, edit and sophisticated the manuscript.
Z.L. helped with the final revision.

**Competing interests**
All contributors declare no competing interests in this work.

**Acknowledgements**
The authors thank Samuel Morin and Suryanarayanan Balasubramanian for their
comments which are crucial for improving this work. This research is supported by
the Second Tibetan Plateau Scientific Expedition and Research (STEP) program
(2019QZKK0201), the Strategic Priority Research Program of the Chinese Academy
of Sciences  (XDA20040501, XDA20060201), the National Natural Science
Foundation of China (41771081), the State Key Laboratory of Cryospheric Sciences
(SKLCS-ZZ-2019) and the Key Research Program of Frontier Sciences of Chinese
Academy of Sciences (QYZDB-SSW-SYS024).

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
