# Peer review of "Applying artificial snowfalls to reduce the melting of the Muz Taw Glacier, Sawir Mountains"

_The Cryosphere, 2019_

## Referee Comment (RC1) · Suryanarayanan Balasubramanian (Referee) · 4 Feb 2020

General Comments This paper is well written and provides good scientific evidence on the impact of artificial precipitation on glacial mass balance. Although, the experiment setup being novel, requires further context in the paper. Even though the paper provides compelling evidence by quantifying the impact of 2 artificial precipitation events, the 13 day measurement duration is too short to provide sufficient evidence for the hypothesis suggested.

Specific Comments 1. Given that the premise of the paper is to measure the effect of artificial precipitation, little effort has been taken to distinguish or categorize precipita-

tion events as artificial and natural. There needs to be a control experiment without igniting the smog generators to compare the difference in precipitation quantities. References are also lacking to categorize the precipitation events as "artificial".

2. The albedo decay of the artificial precipitation and the snow quality data is required to claim a long term glacier mass balance impact. These need to be factored in the hypothesis mechanism. Particularly, the variation in likelihood of a precipitation event occuring with or without a smog generator needs to be quantified or referenced.

Technical Corrections 1. Lines 126 to 137 which describe the AWS instruments can be better presented in the form of a table.

---

## Referee Comment (RC2) · Samuel Morin (Referee) · 4 Feb 2020

Review of « Applying artificial precipitations to mitigate the melting of the Muz Taw Glacier, Sawir Mountains », by Wang et al., by Samuel Morin, 4 February, 2020.

In their manuscript entitled "Applying artificial precipitations to mitigate the melting of the Muz Taw Glacier, Sawir Mountains", Wang et al. report on an experiment where artificial precipitation was produced downstream a mountain glacier in Northern China, and lead to accumulation on the glacier above. The results are discussed in the context of how artificial precipitation could be used to reduce the pace of glacier melt in the context of ongoing climate change. Artificial modifications of the functioning of mountain glaciers is an emerging field, contributing to a larger move of the scientific community towards assessing the potential of geoengineering – which proceeds through various mechanisms and approaches – to reduce the magnitude and impact of climate change at various time scales. Such studies are probably unavoidable, and they are rendered necessary by the push from some societal compartments to apply geoengineering, there is thus a need to carefully assess the impacts, implications, potentials benefits and risks, of such approaches, and this study contributes to this activity. Overall, I think that the data acquired for this study are appropriate to address whether artificial precipitation has a significant impact – or not, on glacier mass balance, but the manuscript suffers from many shortcomings (including a general lack of clarity in how the results are presented and the data compared and interpreted), which I hope that the authors can address before the manuscript can be recommended for publication. I have several major concerns, see below, and series of other editorial comments and suggestions.

Major concerns

Reduction in mass loss: For this study, it seems that the artificial precipitation was applied in summertime, at time of glacier ablation and melt (August 2018). However, it is unclear, whether the decrease in mass loss, reported to be 17% in the abstract, accounts for the amount of precipitation added by the artificial precipitation, or not. Indeed, by adding mass to the glacier, the mass lass can only by lower than without artificial precipitation. The impact can be considered significant if the reduction in mass loss exceeds the gain corresponding to the deposition of artificial precipitation. I think this should be clarified.

Environmental footprint of artificial precipitation: It is absolutely necessary that geoengineering methods, applied at various scales, undergo an assessment of their effectiveness and potential side effects. Even if a full assessment of the potential side effect of artificial precipitation may fall beyond the scope of this manuscript, I think that it would be worth mentioning that this is a requirement to be undertaken if this experiment is to be repeated or scaled up. In particular, it would be interesting to be able to know, from reading the article, why is artificial precipitation implemented in these valleys (what is the context for setting up these artificial precipitation units ?), what is the energy and water cost associated to these activities, and, therefore, move towards an attempt to quantify the cost and benefit of the method, i.e. contrast the avoided glacier mass loss with the corresponding effort to reach this goal. I think this it is absolutely necessary that side effects and environmental and economic costs associated to this approach, are mentioned, and even better, quantified in a revised version of the manuscript.

Mechanism : I have major reservations about some aspects of the "possible mechanism" introduced by the authors. It seems clear for me that by adding artificial precipitation, in the form of snow, the albedo of the surface increases, without invoking the influence of cloud cover on surface albedo. See detailed comments below.

Minor comments and suggestions

Title : I think the use of the term "mitigate" in the title of the manuscript is misleading. I think "litigate" could be replaced by "reduce". Mitigation generally refers, in climate change studies, to the reduction in greenhouse gas emissions, which is not the scope of this manuscript.

Page 1, Line 17 : Replace "Glaciers" by "glaciers"

Page 1, Line 18 : after "higher latitude and lower elevations", a qualifier is missing after adding "than", or the sentence needs to be rephrased.

Page 1, Line 20 : replace "in presence" by "observed"

Page 1, Line 21 : add "additional" or "artificial" before "precipitation"

Page 1, Line 24 : replace "MB" by "Mass Balance"

Page 1, Line 25 ; delete "AWS", no need to introduce acronyms in the abstract.

Page 1, Line 26 : delete "EL", no need to introduce acronyms in the abstract.

Page 1, Line 29 : I suggest "decreased by 17%" is clarified, as indicated in my major comment. Also, it should be made more explicit what is the time scale over which the mass balance values are compared. At present, it is unclear whether the reduction applies to annual, monthly, weekly etc. mass balance values.

Page 1, Line 30 : I suggest rephrasing the "possible mechanism" and replacing it with a more concrete statement about the mechanism, see below for further comments on the mechanism as it is introduced in this manuscript.

Page 1, Line 34: I suggest replacing « MB » by « Glacier mass balance » in the keywords. « Melting mitigation » does not seem a fully appropriate keyword (see above).

Page 2, line 37 : Immerzeel et al. (2010) is a solid reference, but there have been more recent and exhaustive and compelling studies published recently on this topic (e.g. Immerzeel et al., 2010, in press, https://doi.org/10.1038/s41586-019-1822-y).

Page 2, line 42 : same here, Zemp et al. (2015) could be replaced by Zemp et al. (2019) for a more up-to-date introduction.

Page 2, line 43 : « more intense » : this needs clarification, currently the text does not state than what the ablation is more intense.

Page 2, line 43 and 44 : total glacier length and total glacier surface are should be provided, and not only the change, so as to provide better context.

Page 2, lines 45 to 49 : Thess sentences are not supported by references ; maybe refer to the Hock et al. IPCC SROCC Chapter (in press) ?

Page 2, lines 51 to 59. I think this pararaph requires major clarifications. First of all, starting on the first sentence, there are not so many approaches used in practice for reducing the rate of glacier ablation. Covering glaciers with insulating material has been described in detail by Fischer et al. (The Cryosphere, 2016), I think it's finding should be quoted in this paper. Also, it is surprising to see « scientists and governments » together acting on « taking measures », and later on, on page 59, that « scientists plan to use artificial snow ». In fact, scientists can assess the impact of various approaches, but I don't think that it can be stated that scientists are « planning » or « taking measures » to reduce glacier mass loss. I think this paragraph should be clarified, in order to better position the respective role of scientists and governing bodies (at local or national scale). I also think that, if the term « geoengineering » is retained (line 55), a definition should be provided, in order to frame this particular article within the climate change geoengineering literature.

Page 2, line 62 tp 63 : it should be made clear whether the artificial precipitation devices were installed on purpose for this particular study, or not, and if this is the case, what is the motivation for installing these equipments in a broader context. Maybe, some more context statements should be given about artificial precipitation technology, its typical context and scope, and why it is potentially interesting to apply it for attempting to reduce glacier mass loss.

Page 3, line 88 : The first statement needs a reference.

Page 3, line 91 : add « surface » before « previous » and « area ».

Page 3, line 92 : I strongly suggest not using acronyms such as « MB ». It does not save much space, and leads to poorer readability.

Page 4, line 93 : It is very unclear what the values « -975 ~ -1286 mm w.e. » mean. Are these annual mass balance values ? What is the range corresponding to ? Is this an uncertainty on glacier-averaged values ? Or a range representing the spatial variability on the glacier ? This should be rephrased for better clarity.

Page 4, line 106 : « When we realized » : this needs to be clarified

Page 4, line 107 : « 14 silver-iodide smog generators » : again, it would be useful to know whether this is the usual purpose of such generators ? Or whether they were installed for other purposes ? This could be added to the introduction, but more technical details can also be provided here.

Page 4, line 109 : is « AP » representing « artificial precipitation » ? If so, I strongly suggest that the plain words are used, and not the acronym. This can be applied throughout the entire manuscript (including figure captions).

Page 6, line 135 : suggestion to replace « the accuracy » by « an accuracy »

Page 6, line 136 : « CR6 » is not very informative. Maybe better to either provide more information to identify the data logger, or drop the information if it is not critically important.

Page 7, line 157 to 164 : I couldn't find if an average value for broadband albedo was computed for the entire glacier, or not. If so, then the method used should be provided.

Page 7, line 166 : I strongly suggest replacing « MB » by « mass balance ».

Page 8, line 184 ; I suggest starting this paragraph with several sentences providing more background about the meteorological conditions during the experiment, in particular on what days there was some natural precipitation (or not). It should also be provided, whether it is expected that the intensity of the melt would be the same before and after the days when artificial precipitation was applied (in order to make the comparison meaningful).

Page 8, lines 200 to 202 : this sentence is very hard to understand, I suggest it is revised for better clarity.

Page 10, line 233 : the use of the symbol « ~ » is deprecated, I suggest using a more appropriate symbol (or use « approx. » for example).

Page 10, line 233 : even though it was stated earlier that mass balance measurements are taken since August 12, I think this should be mentioned along with the values provided, for better clarity, and perhaps provided in mm w.e. per day. It is unclear, in the context, what it means « -300 mm w.e. to -100 mm w.e. after the artificial precipitation » : are the values reset on August 18 ? This is hard to follow. Maybe a table with the mass balance values for various locations, and average over the glacier, and corresponding degree day sums, could help provide a less ambiguous description of the data.

Page 10, line 236. « The APs gained the mass » : this needs revision, it is not clear.

Page 10, line 242 : add « in °C » after « temperature »

Page 10, lines 241 to 250 : Although this is where the key results are provided, it is unclear. I understand that the sum of positive degree days is provided for the two periods before and after the artificial precipitation, along with the mass balance for the entire glacier. To me, this is not enough to assess the efficiency of the artificial precipitation process. Indeed, to provide a more informative comparison, I believe that the authors could compare the simulated melt rare (or mass balance) during the period after artificial precipitation, and compare this value with the value measured, accounting for artificial precipitation. This comparison should also explicitly account for the amount of snow added through the artificial precipitation, because adding snow precipitation can indeed only increase the mass. At present, there is no evidence that adding more precipitation leads to lesser mass loss, specifically. This needs to be analyzed in a more in-depth manner, I think. I also think that it would be critical, if the information can be made available, what is the actual deposition rate due to artificial precipitation, on the glacier. With this data at hand, I believe that the authors could make a more compelling case.

Page 11, Table 1 : This table could fill the gap indicated above, but it does not provide sufficiently clear information. One single albedo value is given. Is this an average over the glacier ? If so, what is the methodology? Same for the mass balance. Is the value applicable since August 12 in both cases, or only applies to the time periods t1 and t2 ? I also don't understand the precipitation value. It seems that natural precipitation occurred during t1. If so, how is it possible to assess the impact of artificial precipitation during t2 ? Only some modelling could be used, I think, to assess the impact of artificial precipitation.

Page 11, line 259 to Page 11, line 285. The entire section 4.4 is very confusing, and I recommend that more work is spent on revising it in light of available scientific evidence. It is quite obvious that adding artificial solid precipitation (snow) to a glacier will (1) increase the mass and (2) increase the albedo. There is no need to develop a theory about this. Adding rain may increase the mass. I doubt that the influence of clouds on snow albedo plays a major rôle here (clouds drastically reduce incoming shortwave radiation, which is the #1 factor most certainly in this case). I suggest that this section should be considerably simplified. Instead of these questionable speculations, I encourage the authors to perform some simple mass balance modeling (e.g. based on degree days values), in order to contrast the mass loss values with and without artificial precipitation. This would make the case more compelling and its results could be more useful to the scientific community.

Page 13, line 292 : I understand that in some parts of the glacier, artificial precipitation did not fall as snow but rather rain. Could this be clarified ? Here we have the impression that artificial precipitation lead to snow precipitation everywhere on the glacier.

Page 13, lines 296 to 303 : this is very confusing. I don't understand what numbers are compared to what, for what periods of time, and what conclusions could be made. I suggest making a thorough revision of this part, because it affects how the efficiency of the artificial precipitation approach can be computed. I strongly suggest making comparisons pertaining to the same time periods, and not comparing different time periods. Again, modelling could be used to place the artificial precipitation experiment in a clearer context.

Page 13, line 305 to 311 : see above my comments about the « physical mechanism ». I think much simpler statements are sufficient to explain the observations. However, as indicated in my major comments, I think that the reader expects, at the end of the conclusion, a broader perspective on this work, a discussion on the efficiency of this « geoengineering » approach (including an assessment of the energy costs for artificial precipitation, to be compared to the benefit of reducing mass loss). It could also be discussed whether the authors have recommendations on future research, in particular

in the (possible) context where such a method could be implemented at a wider scale or more regularly. All these questions should be at least mentioned by the authors.

Figures :

Figure 2 : replace « Ladar » by « Radar »

Figure 4 : onset picture is not readable. If the content is useful to the reader, then it should be provided as clearly readable image. Also, what is « contour line » as indicated in the legend ? I also couldn't find the « equilibrium line » on the figure, because several lines have almost the same style. Some editing is required.

Figure 5 : I suggest adding vertical shaded areas to indicate the periods when artificial precipitation was applied. Also, the figure quality should be improved, on the pdf provided for review the image quality is quite bad.

Figure 6 : the albedo values in the various onset figures is very hard to read. I suggest using a more classical design, with numbers referring to the measurement sites, and larger plots on the side of the map. The information will be better conveyed.

Figure 7 : this figure is very confusing. Is « gained mass » the direct consequence of artificial precipitation ? Or is it the difference between the two « mass balance » time series (which is confusing, because it is indicated that the reference is on August 12 for all values), which would then combine not only artificial precipitation but also melt after the precipitation. Better clarity and, probably better language to describe what is displayed on the graphs, are needed.

---

## Author Response (AR1)

**AC1\_Reply to the comments of Suryanarayanan Balasubramanian**

- 23 General Comments This paper is well written and provides good scientific evidence on the
  - 4 impact of artificial precipitation on glacial mass balance. Although, the experiment setup
  - 5 being novel, requires further context in the paper. Even though the paper provides
  - 6 compelling evidence by quantifying the impact of 2 artificial precipitation events, the 13-day
  - 7 measurement duration is too short to provide sufficient evidence for the hypothesis8 suggested.
  - 9 *Re:* We thank the referee for the valuable comments which are believed to be greatly
- 10 helpful for improving the quality of the manuscript. The work itself is preliminary and needs
- 11 more data to consolidate the current knowledge in future. We have plan to apply for
- 12 funding for carrying out more intensive experiments in this glacier and/or other glaciers. We
- 13 also include these concerns in the revised manuscript.
- 14
- 15 Specific Comments
- 16 1. Given that the premise of the paper is to measure the effect of artificial precipitation,
- 17 little effort has been taken to distinguish or categorize precipitation events as artificial and
- 18 natural. There needs to be a control experiment without igniting the smog generators to
- 19 compare the difference in precipitation quantities. References are also lacking to categorize
- 20 the precipitation events as "artificial".
- 21 *Re:* Thanks. We added the description including a new figure on how we operated the AgI
- 22 smoke generators and when the AWS recorded the consequent snowfalls in the revised
- 23 manuscript. There were significant snowfall amounts recorded by the AWS every single time
- 24 after we ignited the smoke generators. We could not completely distinguish the artificial
- 25 snowfalls from the natural one if they were mixed in all these events. However, the co-
- 26 occurring of the snow falling with the AgI smoke allows us to affirm that we were producing
- 27 some artificial snowfall. The reply has been integrated into and underlined in the revised
- 28 manuscript.
- 29
- 30 2. The albedo decay of the artificial precipitation and the snow quality data is required to
- 31 claim a long-term glacier mass balance impact. These need to be factored in the hypothesis
- 32 mechanism. Particularly, the variation in likelihood of a precipitation event occurring with or
- 33 without a smog generator needs to be quantified or referenced.
- 34 *Re:* Yes, thanks. These concerns have been added into the relevant context. As we
- 35 addressed in the aforementioned reply, this is a very preliminary experiment and need
- 36 further studies to validate our method and theory. However, we include these new
- 37 ingredients in our revised manuscript.
- 38
- 39 Technical Corrections
- 40 1. Lines 126 to 137 which describe the AWS instruments can be better presented in the
- 41 form of a table.
- 42 *Re:* Thanks. We made a new table (Table 1) for the advice.

**AC2\_Reply to the comments of Samuel Morin**

1 2

3 In their manuscript entitled "Applying artificial precipitations to mitigate the melting of the 4 Muz Taw Glacier, Sawir Mountains", Wang et al. report on an experiment where artificial 5 precipitation was produced downstream a mountain glacier in Northern China, and lead to 6 accumulation on the glacier above. The results are discussed in the context of how artificial 7 precipitation could be used to reduce the pace of glacier melt in the context of ongoing 8 climate change. Artificial modifications of the functioning of mountain glaciers is an 9 emerging field, contributing to a larger move of the scientific community towards assessing 10 the potential of geoengineering - which proceeds through various mechanisms and 11 approaches – to reduce the magnitude and impact of climate change at various time scales. 12 Such studies are probably unavoidable, and they are rendered necessary by the push from 13 some societal compartments to apply geoengineering, there is thus a need to carefully 14 assess the impacts, implications, potentials benefits and risks, of such approaches, and this 15 study contributes to this activity. Overall, I think that the data acquired for this study are 16 appropriate to address whether artificial precipitation has a significant impact – or not, on 17 glacier mass balance, but the manuscript suffers from many shortcomings (including a 18 general lack of clarity in how the results are presented and the data compared and 19 interpreted), which I hope that the authors can address before the manuscript can be 20 recommended for publication. I have several major concerns, see below, and series of other 21 editorial comments and suggestions. 22 Re: The authors thank the reviewer for describing the general impression on our manuscript 23 here. We will address our corrections and improvements in the replies to the specific 24 comments. 25 26 Major concerns 27 Reduction in mass loss: For this study, it seems that the artificial precipitation was applied in 28 summertime, at time of glacier ablation and melt (August 2018). However, it is unclear,

whether the decrease in mass loss, reported to be 17% in the abstract, accounts for the
amount of precipitation added by the artificial precipitation, or not. Indeed, by adding mass
to the glacier, the mass lass can only by lower than without artificial precipitation. The

- 32 impact can be considered significant if the reduction in mass loss exceeds the gain
- 33 corresponding to the deposition of artificial precipitation. I think this should be clarified.
- 34 *Re:* I think there was misunderstanding in the statement of the original abstract. We would
- 35 like to express that "the average mass loss decreased by 41 mm w.e. during and after the
- APs (i.e. 18 24 Aug), accounting for 17% of the mass loss prior to the APs (i.e. 12 18
- Aug)". We rephrased the sentence and underlined it in the revision. In the revised
- 38 manuscript, we made two comparisons separately. One is the aforementioned, and the
- 39 other is comparing the snowfall recorded by the AWS due to the experiments with the total
- 40 melt after the experiments.
- 41
- 42 Environmental footprint of artificial precipitation: It is absolutely necessary that
- 43 geoengineering methods, applied at various scales, undergo an assessment of their
- 44 effectiveness and potential side effects. Even if a full assessment of the potential side effect
- 45 of artificial precipitation may fall beyond the scope of this manuscript, I think that it would
- 46 be worth mentioning that this is a requirement to be undertaken if this experiment is to be
- 47 repeated or scaled up. In particular, it would be interesting to be able to know, from reading

- 1 the article, why is artificial precipitation implemented in these valleys (what is the context
- 2 for setting up these artificial precipitation units?), what is the energy and water cost
- 3 associated to these activities, and, therefore, move towards an attempt to quantify the cost
- 4 and benefit of the method, i.e. contrast the avoided glacier mass loss with the
- 5 corresponding effort to reach this goal. I think this it is absolutely necessary that side effects
- 6 and environmental and economic costs associated to this approach, are mentioned, and
- 7 even better, quantified in a revised version of the manuscript.
- 8 *Re:* Yes, the comment arises an important issue which was not mentioned by the original
- 9 manuscript. We added some text with references in the revision to address the comment.
- 10 The environmental side effects are very low according to a review report released by the
- 11 WMO in 2018. The power used in the smog generators is solar and no extra water is costed.
- 12 The valley-developed glaciers are ideal sites to perform the experiment due to the
- 13 prevailing winds helping carry the smog up over the glacier surface. We have plans to scale
- 14 up the present study to other glaciers in future. These concerns have been integrated into
- 15 the introduction and conclusion parts of the revision and underlined.
- 16
- 17 Mechanism: I have major reservations about some aspects of the "possible mechanism"
- 18 introduced by the authors. It seems clear for me that by adding artificial precipitation, in the
- 19 form of snow, the albedo of the surface increases, without invoking the influence of cloud
- 20 cover on surface albedo. See detailed comments below.
- 21 *Re:* Yes, this part has been significantly simplified according to the specific comment in
- below. We only keep the concern of snowfall increasing mass and albedo mass balance
- part. We exclude Figure 8 from the manuscript.
- 25 Minor comments and suggestions
- 26 Title: I think the use of the term "mitigate" in the title of the manuscript is misleading. I
- 27 think "litigate" could be replaced by "reduce". Mitigation generally refers, in climate change
- studies, to the reduction in greenhouse gas emissions, which is not the scope of thismanuscript.
- 30 *Re:* We did the replacement to the title as advised by the reviewer.
- 31
- 32 Page 1, Line 17: Replace "Glaciers" by "glaciers"
- 33 *Re:* We replaced "Glaciers" by "glaciers" in Line 17.
- 34
- Page 1, Line 18: after "higher latitude and lower elevations", a qualifier is missing after
- 36 adding "than", or the sentence needs to be rephrased.
- *Re:* We add "than those in the adjacent areas" after "higher latitude and lower elevations".
- 39 Page 1, Line 20: replace "in presence" by "observed"
- 40 *Re:* Replaced as advised.
- 41
- 42 Page 1, Line 21: add "additional" or "artificial" before "precipitation"
- 43 *Re:* Yes, we added "artificial".
- 44
- 45 Page 1, Line 24: replace "MB" by "Mass Balance"
- 46 *Re:* We replaced "MB" by "mass balance".
- 47

- 1 Page 1, Line 25: delete "AWS", no need to introduce acronyms in the abstract. Page 1, Line
- 2 26 : delete "EL", no need to introduce acronyms in the abstract.
- 3 *Re:* We deleted them in the revision.
- 4
- 5 Page 1, Line 29: I suggest "decreased by 17%" is clarified, as indicated in my major
- 6 comment. Also, it should be made more explicit what is the time scale over which the mass
- 7 balance values are compared. At present, it is unclear whether the reduction applies to
- 8 annual, monthly, weekly etc. mass balance values.
- 9 *Re:* Yes, we clarified the statement in the abstract and the method. The stick scales for
- 10 measuring mass balance was read thrice, on 12, 18 and 24 Aug, respectively. We compared
- 11 the mass varying between the two periods (12-18 Aug and 18-24 Aug). These have been 12 clarified in the revision.
- 13
- 14 Page 1, Line 30: I suggest rephrasing the "possible mechanism" and replacing it with a more
- 15 concrete statement about the mechanism, see below for further comments on the 16 mechanism as it is introduced in this manuscript
- 16 mechanism as it is introduced in this manuscript.
- 17 *Re:* Yes, we rephrased it and simplified the mechanism part in the revision. We included
- 18 more discussion in the reply to the following comments.
- 19

- 20 Page 1, Line 34: I suggest replacing « MB » by « Glacier mass balance » in the keywords. «
- 21 Melting mitigation » does not seem a fully appropriate keyword (see above).
- 22 *Re:* Yes, we replaced the keywords as suggested.
- Page 2, line 37: Immerzeel et al. (2010) is a solid reference, but there have been more
- 25 recent and exhaustive and compelling studies published recently on this topic (e.g.
- 26 Immerzeel et al., 2010, in press, https://doi.org/10.1038/s41586-019-1822-y).
- 27 *Re:* We added the new reference into the revised.
- 28
- Page 2, line 42 : same here, Zemp et al. (2015) could be replaced by Zemp et al. (2019) for a
  more up-to-date introduction.
- 31 *Re:* We replaced the old literature with the new one.
- 32

Page 2, line 43 : « more intense » : this needs clarification, currently the text does not state
than what the ablation is more intense.

- 35 *Re:* Yes, clarified. "For the Sawir Mountains, the ablation of the glaciers is more intense than
- the global average, and the total area of the glaciers reduced by 46% from 23 km2 in 1977
- 37 to 12.5 km2 in 2017 (Wang et al., 2019)".
- 38
- Page 2, line 43 and 44: total glacier length and total glacier surface are should be provided,
- 40 and not only the change, so as to provide better context.
- 41 *Re:* Yes, the information provided in the revision. "For the Sawir Mountains, the ablation of
- 42 the glaciers is more intense than the global average, and the total area of the glaciers
- 43 reduced by 46% from 23 km2 in 1977 to 12.5 km2 in 2017 (Wang et al., 2019)".
- 44
- 45 Page 2, lines 45 to 49: Thess sentences are not supported by references; maybe refer to the
- 46 Hock et al. IPCC SROCC Chapter (in press)?

*Re:* The advised reference was added into the revision. "The accelerated retreat of glaciers
not only causes spatial and temporal changes in water resources but also has a significant
impact on sea-level rise, regional water cycles, ecosystems and socio-economic systems
(such as agriculture, hydropower and tourism); the melting of glaciers also increases the
occurrence of glacial disasters, such as glacial lake outburst flooding, icefalls and glacial

- 6 debris flows (Hock et al., 2019)".
- 7

8 Page 2, lines 51 to 59. I think this paragraph requires major clarifications. First of all, starting

9 on the first sentence, there are not so many approaches used in practice for reducing the
10 rate of glacier ablation. Covering glaciers with insulating material has been described in

11 detail by Fischer et al. (The Cryosphere, 2016), I think it's finding should be quoted in this

12 paper. Also, it is surprising to see « scientists and governments » together acting on « taking

13 measures », and later on, on page 59, that « scientists plan to use artificial snow ». In fact,

scientists can assess the impact of various approaches, but I don't think that it can be stated

15 that scientists are « planning » or « taking measures » to reduce glacier mass loss. I think

- 16 this paragraph should be clarified, in order to better position the respective role of scientists
- 17 and governing bodies (at local or national scale). I also think that, if the term «
- geoengineering » is retained (line 55), a definition should be provided, in order to frame this
  particular article within the climate change geoengineering literature.
- 20 *Re:* Yes, we rephrased the paragraph. The item "geoengineering" was removed from the

21 original manuscript for the small scale of the study against the definition of the word. We

- 22 clarified the statements involving the roles played by scientists and governments. The
- reference of Fischer et al. (2016) was added into the revision.
- 24

Page 2, line 62 to 63: it should be made clear whether the artificial precipitation devices

26 were installed on purpose for this particular study, or not, and if this is the case, what is the

27 motivation for installing these equipments in a broader context. Maybe, some more context

28 statements should be given about artificial precipitation technology, its typical context and

scope, and why it is potentially interesting to apply it for attempting to reduce glacier massloss.

31 *Re:* We addressed their purpose in the revision. "These smog generators were set up there

32 by the local meteorological service for artificial-precipitation tasks". Some more technic

33 features of these generators are included in the experiment section.

34

35 Page 3, line 88 : The first statement needs a reference.

*Re:* We added a reference. "The Muz Taw Glacier has been in constant recession since 1959
(Wang et al., 2019)".

38

39 Page 3, line 91 : add « surface » before « previous » and « area ».

40 *Re:* We added.

41

Page 3, line 92 : I strongly suggest not using acronyms such as « MB ». It does not save much
space, and leads to poorer readability.

44 *Re:* We replaced the acronyms, MB and AP with their full-length glossaries throughout the

45 manuscript.

- Page 4, line 93: It is very unclear what the values « -975 ~ -1286 mm w.e. » mean. Are these
  annual mass balance values ? What is the range corresponding to ? Is this an uncertainty on
  glacier- averaged values ? Or a range representing the spatial variability on the glacier? This
- 4 should be rephrased for better clarity.
- 5 *Re:* We clarified the mass balance of the glacier measured in separate years in the revised 6 manuscript.
- 7
- 8 Page 4, line 106 : « When we realized » : this needs to be clarified
- 9 *Re:* We monitored the distribution and structural developing of clouds and identified the
- 10 orientation, height and distance of the clouds approaching the glacier at the radar station.
- 11 Associated with observing the moving of the potentially target clouds and the receiving of
- 12 the reflection of the radar transmission, we ignited the smog generators for seeding
- 13 artificial precipitations, when we realized the possibility is high enough to potentially form
- 14 precipitation (Figure 2). The detailed operation of conducting artificial precipitations in the
- 15 study glacier has been described in Xu et al. (2017).
- 16
- Page 4, line 107 : « 14 silver-iodide smog generators » : again, it would be useful to know
  whether this is the usual purpose of such generators ? Or whether they were installed for
- 19 other purposes ? This could be added to the introduction, but more technical details can
- 20 also be provided here.
- *Re:* This purpose of the generators has been included in the revision and addressed in the
   reply to the aforementioned comment.
- 23
- Page 4, line 109 : is « AP » representing « artificial precipitation » ? If so, I strongly suggest
- that the plain words are used, and not the acronym. This can be applied throughout theentire manuscript (including figure captions).
- 27 *Re:* Corrected as advised.
- 28
- 29 Page 6, line 135 : suggestion to replace « the accuracy » by « an accuracy »
- 30 *Re:* Corrected as advised.
- 31
- 32 Page 6, line 136 : « CR6 » is not very informative. Maybe better to either provide more
- information to identify the data logger, or drop the information if it is not critically
   important
- 34 important.
- 35 *Re:* Yes, we supplemented some more relevant information about CR6.
- 36
- 37 Page 7, line 157 to 164 : I couldn't find if an average value for broadband albedo was
- 38 computed for the entire glacier, or not. If so, then the method used should be provided.
- 39 Re: We averaged the broadband albedo based on the site measurements representing an
- 40 average for the entire glacier. We clarified the statement in the revision.
- 41
- 42 Page 7, line 166: I strongly suggest replacing « MB » by « mass balance ».
- 43 *Re:* Corrected as advised.
- 44
- 45 Page 8, line 184: I suggest starting this paragraph with several sentences providing more
- 46 background about the meteorological conditions during the experiment, in particular on
- 47 what days there was some natural precipitation (or not). It should also be provided,

- 1 whether it is expected that the intensity of the melt would be the same before and after the
- 2 days when artificial precipitation was applied (in order to make the comparison
- 3 meaningful).
- 4 *Re:* There are some added text (underlined) in the revised manuscript.
- 5 There was some natural precipitation during 12 14 August, while except this and 6 that in the experiment days, the whole period of 12 – 24 August were sparse in
- 6 that in the experiment days, the whole period of 12 24 August were sparse in
  7 precipitation.
- 8 We could not completely distinguish the artificial snowfalls from the natural ones if 9 they were simultaneously mixed in all these events. However, the co-occurring of 10 the significantly snow falling with the AgI smoke allows to suppose that we were
- 11 producing artificial snowfalls.
- 12
- Page 8, lines 200 to 202: this sentence is very hard to understand, I suggest it is revised forbetter clarity.
- 15 *Re:* We replaced this statement by "We would compare the intensity of the melt would be
- 16 the same or not before and after the days when artificial precipitation was applied".
- 17
- 18 Page 10, line 233: the use of the symbol « ~ » is deprecated, I suggest using a more
- 19 appropriate symbol (or use « approx. » for example).
- 20 *Re:* Corrected as advised.
- 21

Page 10, line 233: even though it was stated earlier that mass balance measurements are

- taken since August 12, I think this should be mentioned along with the values provided, for
- better clarity, and perhaps provided in mm w.e. per day. It is unclear, in the context, what it
   means « -300 mm w.e. to 100 mm w.e. after the artificial precipitation » : are the values
- reset on August 18 ? This is hard to follow. Maybe a table with the mass balance values for
- various locations, and average over the glacier, and corresponding degree day sums, could
   beln provide a less ambiguous description of the data
- 28 help provide a less ambiguous description of the data.
- 29 *Re:* We only have three readings from the scales of the stakes, which were read on 12, 18
- and 24 August, respectively (Section 3.4). To study the effects of the artificial precipitations
   on the mass balance of the glacier, we calculated the mass balance measured by the stakes
- during the two periods, i.e. 12 18 Aug and 18 24 Aug, respectively. We do not have the
- 33 data for mass balance on a daily basis.
- 34
- Page 10, line 236. « The APs gained the mass » : this needs revision, it is not clear. Page 10,
- 36 line 242 : add « in °C » after « temperature »
- 37 *Re:* Yes, revised as per the advice.
- 38

Page 10, lines 241 to 250: Although this is where the key results are provided, it is unclear. I
understand that the sum of positive degree days is provided for the two periods before and
after the artificial precipitation, along with the mass balance for the entire glacier. To me,

42 this is not enough to assess the efficiency of the artificial precipitation process. Indeed, to

- 43 provide a more informative comparison, I believe that the authors could compare the
- 44 simulated melt rare (or mass balance) during the period after artificial precipitation, and
- 45 compare this value with the value measured, accounting for artificial precipitation. This
- 46 comparison should also explicitly account for the amount of snow added through the
- 47 artificial precipitation, because adding snow precipitation can indeed only increase the

1 mass. At present, there is no evidence that adding more precipitation leads to lesser mass 2 loss, specifically. This needs to be analyzed in a more in-depth manner, I think. I also think 3 that it would be critical, if the information can be made available, what is the actual 4 deposition rate due to artificial precipitation, on the glacier. With this data at hand, I believe 5 that the authors could make a more compelling case. 6 Re: Yes, we added some further analysis. The accumulation at the equilibrium line altitude 7 (ELA) of a glacier is approximately equal to the area average of accumulation over the whole glacier (Braithwaite, 2008). We can presume that the snowfall amount measured by the 8 9 AWS near the ELA of the Muz Taw glacier during t2 was the average received mass of the 10 whole glacier after implementing the artificial precipitatons. The melt amount from the 11 original glacier during t2 would be the difference between the calculated mass balance and 12 the snowfall measured by the gauge on the AWS, i.e. 17.3 mm w.e. Therefore, artificial 13 precipitations may significantly save the melt of the glacier by 53.5%, simply calculated as 14 the percentage of the snowfall divided by the estimated mass balance during t2. 15 16 Page 11, Table 1: This table could fill the gap indicated above, but it does not provide 17 sufficiently clear information. One single albedo value is given. Is this an average over the

18 glacier? If so, what is the methodology? Same for the mass balance. Is the value applicable

19 since August 12 in both cases, or only applies to the time periods t1 and t2? I also don't

20 understand the precipitation value. It seems that natural precipitation occurred during t1. If

so, how is it possible to assess the impact of artificial precipitation during t2? Only some

22 modelling could be used, I think, to assess the impact of artificial precipitation.

*Re:* Yes, we clarified the content in Table 2 (original Table 1). Please refer to the reply to the
 previous comment and the revised manuscript.

25

26 Page 11, line 259 to Page 11, line 285. The entire section 4.4 is very confusing, and I 27 recommend that more work is spent on revising it in light of available scientific evidence. It 28 is quite obvious that adding artificial solid precipitation (snow) to a glacier will (1) increase 29 the mass and (2) increase the albedo. There is no need to develop a theory about this. 30 Adding rain may increase the mass. I doubt that the influence of clouds on snow albedo 31 plays a major rôle here (clouds drastically reduce incoming shortwave radiation, which is the 32 #1 factor most certainly in this case). I suggest that this section should be considerably simplified. Instead of these questionable speculations, I encourage the authors to perform 33 34 some simple mass balance modelling (e.g. based on degree days values), in order to 35 contrast the mass loss values with and without artificial precipitation. This would make the 36 case more compelling and its results could be more useful to the scientific community. 37 *Re:* We largely simplified this part. We input some new discussion in the last paragraph of 38 Section 4.3, contrasting the mass loss with and without artificial precipitation.

39

Page 13, line 292: I understand that in some parts of the glacier, artificial precipitation did
not fall as snow but rather rain. Could this be clarified? Here we have the impression that

42 artificial precipitation leads to snow precipitation everywhere on the glacier.

43 *Re:* In Section 4.1, we have discussed when the precipitation is snow or rain under some

44 circumstances. In our experiments, the glacier received snow as observed. We clarified the

45 statement in the sentence, avoiding further confusion.

1 Page 13, lines 296 to 303: this is very confusing. I don't understand what numbers are

2 compared to what, for what periods of time, and what conclusions could be made. I suggest

3 making a thorough revision of this part, because it affects how the efficiency of the artificial

4 precipitation approach can be computed. I strongly suggest making comparisons pertaining

- 5 to the same time periods, and not comparing different time periods. Again, modelling could
- 6 be used to place the artificial precipitation experiment in a clearer context.
- 7 *Re:* We did two comparisons for the mass-balance variation of the Muz Taw glacier with or
- 8 without the artificial-snowfall experiments. One is comparing the mass balance during the
- 9 period before the experiments (12 18 Aug) with that after (18 24 Aug). The difference of
- 10 the mass balances between the two periods was  $41 \pm 15$  mm w.e., suggesting that artificial

snow added the mass to the glacier. Another is comparing the total melt of the glacier

- during the period after the experiments (18 24 Aug) with the mass added from the
   artificial snowfall to the glacier, implying that artificial snow significantly saved the mass loss
- 14 during the period after the experiments.
- 15

Page 13, line 305 to 311: see above my comments about the « physical mechanism ». I think
much simpler statements are sufficient to explain the observations. However, as indicated
in my major comments, I think that the reader expects, at the end of the conclusion, a

- 19 broader perspective on this work, a discussion on the efficiency of this « geoengineering »
- 20 approach (including an assessment of the energy costs for artificial precipitation, to be
- 21 compared to the benefit of reducing mass loss). It could also be discussed whether the
- 22 authors have recommendations on future research, in particular in the (possible) context
- 23 where such a method could be implemented at a wider scale or more regularly. All these
- 24 questions should be at least mentioned by the authors.

25 *Re:* As shown in the reply to the previous comments, the mechanism has been largely

simplified. And an additional paragraph has been added into the revised manuscript toaddress the future perspective on the study.

- 28 "The approach in our work uses solar power to ignite the seeding material for forming
- 29 clouds and uses no extra water but redistributes natural water in the local atmosphere at a
- 30 small spatial scale. The energy-and-water saving techniques of the approach with
- 31 reasonably mass-loss-reducing efficiency from the Muz Taw glacier validates its efficiency to
- 32 possibly be applied in more Central-Asian glaciers to reduce their rapid melting. Especially in
- 33 summer when the melting is drastic in the Central-Asian glaciers, applying the approach
- 34 suggested by our study on a much broader scale might reduce the melting significantly. Of
- 35 course, the period of our experiment is preliminary and short, and the approach would
- 36 sophisticate itself when being implemented more regularly in future repeated and longer-
- 37 term, or scaled-up experiments.".
- 38
- 39 Figures:
- 40 Figure 2: replace « Ladar » by « Radar »
- 41 *Re:* Corrected.
- 42
- 43 Figure 4: onset picture is not readable. If the content is useful to the reader, then it should
- 44 be provided as clearly readable image. Also, what is « contour line » as indicated in the
- 45 legend? I also couldn't find the « equilibrium line » on the figure, because several lines have
- 46 almost the same style. Some editing is required.
- 47 *Re:* We have redesigned the figure. The submitted file has a larger resolution.

- 1
- 2 Figure 5: I suggest adding vertical shaded areas to indicate the periods when artificial
- 3 precipitation was applied. Also, the figure quality should be improved, on the pdf provided
- 4 for review the image quality is quite bad.
- 5 *Re:* We improved the quality of Figure 5 as advised by the reviewer.
- 6
- 7 Figure 6: the albedo values in the various onset figures is very hard to read. I suggest using a
- 8 more classical design, with numbers referring to the measurement sites, and larger plots on
- 9 the side of the map. The information will be better conveyed.
- 10 *Re:* Yes, we redesigned the layout of the figure complying with the comment.
- 11
- 12 Figure 7: this figure is very confusing. Is « gained mass » the direct consequence of artificial
- 13 precipitation? Or is it the difference between the two « mass balance » time series (which is
- 14 confusing, because it is indicated that the reference is on August 12 for all values), which
- 15 would then combine not only artificial precipitation but also melt after the precipitation.
- 16 Better clarity and, probably better language to describe what is displayed on the graphs, are
- 17 needed.
- 18 *Re:* Yes, the gained mass meant to be the difference between two periods and has been
- 19 clarified. We have clarified the statement including the text and figure in the revision.
- 20

| 1  | List of major changes  |                                                                                           |  |  |  |
|----|------------------------|-------------------------------------------------------------------------------------------|--|--|--|
| 2  | Changed Points to RC 1 |                                                                                           |  |  |  |
| 3  | 1.                     | We added a new figure (Figure 5b) in the revision to describe the cause-                  |  |  |  |
| 4  |                        | consequence between the AgI smoke and the consequent snowfalls in the revised             |  |  |  |
| 5  |                        | manuscript. There were significant snowfall amounts recorded by the AWS every             |  |  |  |
| 6  |                        | single time after we ignited the smoke generators. We could not completely                |  |  |  |
| 7  |                        | distinguish the artificial snowfalls from the natural one if they were mixed in all       |  |  |  |
| 8  |                        | these events. However, the co-occurring of the snow falling with the AgI smoke            |  |  |  |
| 9  |                        | allows us to affirm that we were producing some artificial snowfall. The change has       |  |  |  |
| 10 |                        | been underlined in the revised manuscript.                                                |  |  |  |
| 11 | 2.                     | The variation in likelihood of a precipitation event occurring with or without a smog     |  |  |  |
| 12 |                        | generator has been referenced in the revision, also supplemented by the Changed           |  |  |  |
| 13 |                        | Point 1. We added additional explanation in the end of Section 4.4, "This is a very       |  |  |  |
| 14 |                        | preliminary theory based on the limited data derived from the short-term                  |  |  |  |
| 15 |                        | experiments, and we need further studies to validate the theory. The albedo decay         |  |  |  |
| 16 |                        | of artificial snowfall and snow physics are required to claim a long-term impact on       |  |  |  |
| 17 |                        | the mass balance of glaciers. Particularly, the variation in the likelihood of a snowfall |  |  |  |
| 18 |                        | event occurring with or without smoke generators needs to be quantified in future         |  |  |  |
| 19 |                        | studies.".                                                                                |  |  |  |
| 20 | 3.                     | A new Table 1 was added into the revision.                                                |  |  |  |
| 21 |                        |                                                                                           |  |  |  |
| 22 |                        |                                                                                           |  |  |  |
| 23 | Changed Points to RC 2 |                                                                                           |  |  |  |
| 24 | 1.                     | We changed the manuscript title from "precipitations" to "snowfalls", in the              |  |  |  |
| 25 |                        | wake of the main purpose of our study to add artificial snow onto glacier and largely     |  |  |  |
| 26 |                        | simplifying the hypothesis excluding rainfall and other cases.                            |  |  |  |
| 27 | 2.                     | In the abstract, we clarified some unclear statements. For example, we would like to      |  |  |  |
| 28 |                        | express that "the average mass loss decreased by 41 mm w.e. during and after the          |  |  |  |
| 29 |                        | APs (i.e. 18 – 24 Aug), accounting for 17% of the mass loss prior to the APs (i.e. 12 –   |  |  |  |
| 30 |                        | 18 Aug)". We rephrased the sentence and underlined it in the revision. In the revised     |  |  |  |
| 31 |                        | manuscript, we made two comparisons separately. One is the aforementioned, and            |  |  |  |
| 32 |                        | the other is comparing the snowfall recorded by the AWS due to the experiments            |  |  |  |

1 with the total melt after the experiments. Some other typo and grammatical errors 2 are also corrected and underlined in the revision. 3 3. New keywords: artificial snowfall, Muz Taw Glacier, Sawir Mountains, glacier mass 4 balance, reduce melting 5 4. We outlook the environmental side effects in the conclusion part, which are very low 6 according to a review report released by the WMO in 2018. The power used in the 7 smog generators is solar and no extra water is costed. The valley-developed glaciers are ideal sites to perform the experiment due to the prevailing winds helping carry 8 9 the smog up over the glacier surface. We have plans to scale up the present study to 10 other glaciers in future. These concerns have been integrated into the introduction 11 and conclusion parts of the revision and underlined. 5. We largely simplified the mechanism part, with only keeping the "snowfall -12 13 increasing mass and albedo – mass balance" part. We also exclude Figure 8 from the 14 manuscript according to the simplification. 15 6. We supplemented some necessary references and integrated some useful information into the revision. For example, 16 17 i. Zemp, M., Huss, M., Thibert, E., Eckert, N., et al.: Global glacier mass changes and their contributions to sea-level rise from 1961 to 2016, 18 19 Nature, 568, 382-386, 2019. 20 ii. Immerzeel, W. W., Lutz, A. F., Andrade, M., Bahl, A., et al.: Importance 21 and vulnerability of the world's water towers, Nature, doi: 22 10.1038/s41586-019-1822-y, 2019. 2019. 23 iii. Hock, R., Rasul, G., Adler, C., et al.: High Mountain Areas. In: IPCC 24 Special Report on the Ocean and Cryosphere in a Changing Climate, 25 IPCC, New York, 2019. 26 iv. Fischer, A., Helfricht, K., and Stocker-Waldhuber, M.: Local reduction 27 of decadal glacier thickness loss through mass balance management in ski resorts, The Cryosphere, 10, 2941-2952, 2016. 28 29 v. Flossmann, A. I., Manton, M. J., Abshaev, A., et al.: Peer Review 30 Report on Global Precipitation Enhancement Activities, 2018. 2018. 31 7. We rephrased the the second paragraph of the Introduction part. The item 32 "geoengineering" was removed from the original manuscript for the small scale of

- the study against the definition of the word. We clarified the statements involving
   the roles played by scientists and governments. The reference of Fischer et al. (2016)
   was added into the revision.
- We addressed their purpose in the revision. "These smog generators were set up
   there by the local meteorological service for artificial-precipitation tasks". Some
   more technic features of these generators are included in the experiment section.
- 9. Some acronyms, such as "MB", "AP", etc. have been changed to their full-length
  glossaries throughout the manuscript.
- 9 10. In the revision, we addressed the time-window capture to operate the artificial precipitation experiments. We monitored the distribution and structural developing
   of clouds and identified the orientation, height and distance of the clouds
   approaching the glacier at the radar station. Associated with observing the moving of
- the potentially target clouds and the receiving of the reflection of the radar
  transmission, we ignited the smog generators for seeding artificial precipitations,
  when we realized the possibility is high enough to potentially form precipitation
  (Figure 2). The detailed operation of conducting artificial precipitations in the study
  glacier has been described in Xu et al. (2017). See from line 125 to 141 in the
- 18 revision.
- 19 11. The background about the meteorological conditions during the experiment, in
   20 particular on what days there was some natural precipitation. There are some added
   21 text (underlined) in the revised manuscript.
- There was some natural precipitation during 12 14 August, while except
   this and that in the experiment days, the whole period of 12 24 August were sparse
   in precipitation.
- We could not completely distinguish the artificial snowfalls from the natural
   ones if they were simultaneously mixed in all these events. However, the co occurring of the significantly snow falling with the AgI smoke allows to suppose that
   we were producing artificial snowfalls.
- 12. In the revised manuscript, we clarified how the mass balance was measured by the
  stick scales at each site and how the comparison was made between different
  periods. We only have three readings from the scales of the stakes, which were read
- 32 on 12, 18 and 24 August, respectively (Section 3.4). To study the effects of the

1 artificial precipitations on the mass balance of the glacier, we calculated the mass 2 balance measured by the stakes during the two periods, i.e. 12 – 18 Aug and 18 – 24 3 Aug, respectively. We do not have the data for mass balance on a daily basis. 4 13. We did two comparisons for the mass-balance variation of the Muz Taw glacier with 5 or without the artificial-snowfall experiments. One is comparing the mass balance 6 during the period before the experiments (12 - 18 Aug) with that after (18 - 24 Aug). 7 The difference of the mass balances between the two periods was  $41 \pm 15$  mm w.e., 8 suggesting that artificial snow added the mass to the glacier. Another is comparing the total melt of the glacier during the period after the experiments (18 - 24 Aug)9 10 with the mass added from the artificial snowfall to the glacier, implying that artificial 11 snow significantly saved the mass loss during the period after the experiments. 12 14. Especially for the side effect and promising perspective of broader application of the 13 artificial snow adding mass to glacier, we have added some text to explain the issue 14 in the revision. See the last paragraph in the conclusion (underlined) in the revision. 15 15. All the issues of the figures has been accordingly addressed in the revision. 16

[revised manuscript text omitted]

---

## Author Response (AR2)

**Response to the comments of Referee #2**

2020-05-02

The authors thank the referee for the crucial but insightful comments on the first revised manuscript. We did a substantial revision on the current manuscript based on the comments. The address to the comments is as follows,

1. We address the controversy on the efficacy and scientific validation of seeding cloud to produce enhanced precipitation in history and state that our work is a preliminary attempt in science and engineering in the Introduction part.

2. We collected the precipitation data from an AWS set up in the forefield of the Muz Taw glacier and clear off the AgI smoke. After comparing the precipitation data of the two AWSs, we estimate that natural precipitation either does not involve in the target glacier or accounts for up to 21% of the total precipitation recorded by the AWS at ELA.

3. The estimations of the role of artificial snow reducing the melt of the Muz Taw glacier are based on the new result in point 2.

4. In Section 4.1, we introduced how to partition natural precipitation from the total recorded by the AWS at ELA in detail and assess the possible portion of natural precipitation accounting for the total.

5. Figures. We re-edited Figure 2 and Figure 4. We added a new sub-plot on the relationship of the timings igniting the AgI bars and starting recording snowfall in Figure 5. We added Figure 6 on comparing the precipitations recorded by the two AWSs and their ratios.

6. The weakness of the current study is pointed out both in the Abstract and Conclusion parts.

7. We rechecked and edited the language.

8. The changes and corrections related to the comments have been underlined with red lines.

---

## Author Response (AR3)

Editor Decision: Publish subject to minor revisions (review by editor) (27 May 2020)

by Xavier Fettweis

Comments to the Author:

Dear Authors,

With respect to the recommendations of the reviewer (who I would like to thank a lot), I'm happy to accept your paper for publication. However, some well justified minor revisions, requested by reviewer, are still needed that I will review afterwards myself.

All the best,

Xavier F.
* * *
Reply to Xavier F.

Dear Editor,

We are happy to know the final results of this manuscript and appreciate the reviewer's and your effort and time. Next you can read our responses to the comments of the reviewer.

Best regards,

Feiteng Wang

On behalf of the coauthorship

**Review of second revision of "Applying artificial snowfalls to reduce the melting of the Muz Taw Glacier, Sawir Mountains" by Wang et al.**
**Samuel Morin, 24 May 2020.**

Wang et al. have performed extensive modifications on their manuscript, in order to address a flaw identified during the previous review round. Indeed, the revised manuscript clearly recognize the difficulty in partitioning natural and artificial precipitation in cloud seeding experiments, and the revised manuscript provides a more balanced overview of this difficulty for the present experiment.

I have no more major reservations on this manuscript, although there are several instances, as indicated before, where I think the manuscript deserves to be improved and made clearer. I expect that the editing process following the potential acceptance of the manuscript will provide an opportunity to clarify the wording and remove potential remaining ambiguities. Please find below some suggestions and comments, which can hopefully be quickly addressed by the authors.

*Re:* We appreciate that the reviewer gives recognition to our revision and will respond to the comments one by one below. The modified places in the manuscript have been underlined with red lines.

Page 2, line 13: The statement "decreased by 32 to 41 mm w.e." can be misleading. I suggest rephrasing into "decreased by between 32 and 41 mm w.e." and propagate this change throughout the manuscript.

*Re:* The confusing statement has been changed and propagated throughout the context.

Page 3, line 41 to 43: the statement about ski resorts near Grenoble is totally extraneous to the topic of this publication, I strongly suggest to remove it, as it makes little sense in the logical flow of information.

*Re:* We have removed it from the manuscript.

Page 3, line 46: The term "administrative" is not appropriate here, I suggest deleting, or replace by a more adequate term ("technical" ?)

*Re:* We have changed the term from "administrative" to "technical".

Page 3, line 57: This paragraph starts abruptly about the toxicity of cloud seeding material, without any introduction about cloud seeding and whether this has been tested for reducing glacier mass loss. I strongly suggest that some statements are added prior to this paragraph, in order to introduce cloud seeding in general, and whether this has been tested for reducing glacier mass loss in previous study (apparently not), and indicate that this is the purpose of this study. The paragraph from Page 3, line 57 to Page 4, line 63, could then be positioned at the end of the paragraph from Page 4, line 65 to Page 4, line 75. All in all, the flow of the information needs to be carefully checked, in order to guide the reader through the introduction in a meaningful way.

*Re:* In the beginning of the paragraph, we added "Cloud-seeding over a glacier to generate and enhance snowfall for reducing mass loss has rarely been tested in previous study". We also re-position the original paragraph from Page 3, line 57 to Page 4, line 63 to Page 4, line 75 in the original text.

Page 6, line 117: The paragraph from Page 6, line 117 to Page 6, line 125, does not specifically belong to the subsection "3.1 Artificial-precipitation experiment" but

rather another subsection to be added before it, which could be referred to as "3.1 Meteorological radar observations".

*Re:* We adjusted the structure as advised.

Page 6, line 128: "We" in "We distributed" is unclear. In previous statements, it was indicated that the smog generators were installed by the local meteorological service (Page 4, line 80 "These smog generators were set up there by the local meteorological service for artificial-precipitation tasks". It becomes unclear whether the smog generators were installed there already, and used by opportunity to test whether they could be efficient in reducing glacier mass loss, are if they were installed specifically for the purpose of the experiment.

*Re:* The statement has been changed to "Fourteen silver-iodide (AgI) smog generators have been distributed along the rivers for artificial-precipitation tasks by the local meteorological service".

Page 12, line 291: I suggest that it is always stated which AWS is dealt with, when referred to in the text. Only referring to "AWS" is unclear because it could refer to the "grassland" or "ELA" AWS. I also suggest, for clarity, that the "ELA AWS" is in fact referred to as the "glacier AWS". I believe this will increase the overall readability.

Re: The AWS denotations throughout the manuscript have been specified and checked for consistency.

Page 16, line 361: I suggest replacing "less" by "lower".
*Re:* Yes, has been replaced.

---

## Author Response (AR4)

Comments to the Author pg2, line 14: units are mmWE/day ? or mmWE/7 days? value in % can also be added in the abstract.

Re: The units are denoting the gross mass losses during two periods. We adjusted the position of "after the experiments (Aug 18 – 24)", expecting to clarify the statement. We added the values in percent in the revision.

pg 11, line 277: If yes, I prefer 35% in the manuscript than 0.35. idem 21 +/-3 %

Re: Have been modified here and also in Figure 6.

pg 10, line 236-237: units of precip are mm/day

Re: Yes, we modified the units.

pg 13, Fig6: units of precip are mm/h and not mm.

Re: Corrected.

pg 15-18 + Fig 8 + Table 2: same remark than for the abstract. mmWE/day ? All the values in mmWE should be checked. mm/day? mm/7days ? mm/hour ?

Re: In this section (pg 15 - 18), the unit mm w.e. is specifically denoting the mass balances during the two periods, one of which is during the week before conducting the artificial precipitation and the other is the week after that. This has been priorly declared in the beginning of the section text, legends and captions in Fig 8 and captions of Table 2. We marked the relevant corrections in the revision.

pg 18, line 440: A sentence like below could be added in the conclusion: "However, it is important to note that our approach needs a priori atmospheric conditions favourable to precipitation and can not be applied if the weather is dry and sunny."

Re: Yes, added.

[revised manuscript text omitted]